# A Systematic Study of the In Vitro Pharmacokinetics and Estimated Human In Vivo Clearance of Indole and Indazole-3-Carboxamide Synthetic Cannabinoid Receptor Agonists Detected on the Illicit Drug Market

**DOI:** 10.3390/molecules26051396

**Published:** 2021-03-05

**Authors:** Andrew M. Brandon, Lysbeth H. Antonides, Jennifer Riley, Ola Epemolu, Denise A. McKeown, Kevin D. Read, Craig McKenzie

**Affiliations:** 1Leverhulme Research Centre for Forensic Science, School of Science and Engineering, University of Dundee, Dundee DD1 4HN, UK; a.m.brandon@dundee.ac.uk (A.M.B.); l.h.antonides@dundee.ac.uk (L.H.A.); 2Drug Discovery Unit, Wellcome Centre for Anti-Infectives Research, School of Life Sciences, University of Dundee, Dundee DD1 5EH, UK; j.y.riley@dundee.ac.uk (J.R.); r.epemolu@dundee.ac.uk (O.E.); 3Forensic Medicine and Science, School of Medicine, Dentistry and Nursing, College of Medical, Veterinary and Life Sciences, University of Glasgow, Glasgow G12 8QQ, UK; Denise.McKeown@glasgow.ac.uk

**Keywords:** new psychoactive substances, synthetic cannabinoid receptor agonists, in vitro metabolism, in vivo prediction, pharmacokinetics

## Abstract

In vitro pharmacokinetic studies were conducted on enantiomer pairs of twelve valinate or *tert*-leucinate indole and indazole-3-carboxamide synthetic cannabinoid receptor agonists (SCRAs) detected on the illicit drug market to investigate their physicochemical parameters and structure-metabolism relationships (SMRs). Experimentally derived Log D_7.4_ ranged from 2.81 (AB-FUBINACA) to 4.95 (MDMB-4en-PINACA) and all SCRAs tested were highly protein bound, ranging from 88.9 ± 0.49% ((*R*)-4F-MDMB-BINACA) to 99.5 ± 0.08% ((*S*)-MDMB-FUBINACA). Most tested SCRAs were cleared rapidly in vitro in pooled human liver microsomes (pHLM) and pooled cryopreserved human hepatocytes (pHHeps). Intrinsic clearance (CL_int_) ranged from 13.7 ± 4.06 ((*R*)-AB-FUBINACA) to 2944 ± 95.9 mL min^−1^ kg^−1^ ((*S*)-AMB-FUBINACA) in pHLM, and from 110 ± 34.5 ((*S*)-AB-FUBINACA) to 3216 ± 607 mL min^−1^ kg^−1^ ((*S*)-AMB-FUBINACA) in pHHeps. Predicted Human in vivo hepatic clearance (CL_H_) ranged from 0.34 ± 0.09 ((*S*)-AB-FUBINACA) to 17.79 ± 0.20 mL min^−1^ kg^−1^ ((*S*)-5F-AMB-PINACA) in pHLM and 1.39 ± 0.27 ((*S*)-MDMB-FUBINACA) to 18.25 ± 0.12 mL min^−1^ kg^−1^ ((*S*)-5F-AMB-PINACA) in pHHeps. Valinate and *tert*-leucinate indole and indazole-3-carboxamide SCRAs are often rapidly metabolised in vitro but are highly protein bound in vivo and therefore predicted in vivo CL_H_ is much slower than CL_int_. This is likely to give rise to longer detection windows of these substances and their metabolites in urine, possibly as a result of accumulation of parent drug in lipid-rich tissues, with redistribution into the circulatory system and subsequent metabolism.

## 1. Introduction

Synthetic cannabinoid receptor agonists (SCRAs) are a diverse group of new psychoactive substances (NPS) that bind to, and activate, human cannabinoid receptors (CB_1_ and CB_2_) [1,2]. More than 200 of these compounds have been reported on the illicit market to date [3,4]. SCRA intoxications have been linked to a variety of adverse effects and have been implicated in drug intoxication and drug death cases worldwide [5,6]. The exact mechanisms of many of their harmful effects remain unclear, although some, at least, are mediated via the CB_1_ receptor [7,8,9,10,11]. The intensity and duration of the physiological and psychoactive effects experienced by users of SCRAs will be influenced by their pharmacodynamics and pharmacokinetics, as well as other factors such as the dose, presence of co-ingested substances and individual factors such as sex and underlying health status [11]. Commonly, the actual drug present and its concentration in a preparation will be unknown to the user. Pharmacokinetics and pharmacodynamics will vary between SCRA structural classes [12,13,14,15,16,17,18,19,20,21], being influenced by physicochemical properties related to structural features.

The chemical structures of the SCRAs discussed in this study, including a representation showing the four constituent structural subunits (head, linker, core, tail), are provided in Figure 1 and are referred to by numbers in bold in parenthesis throughout the text. Amongst the most prevalent and potent SCRAs are the valinate and *tert*-leucinate indole and indazole-3-carboxamides. These include AMB-FUBINACA (**1**) (MMB-FUBINACA; methyl [1-(4-fluorobenzyl)-1*H*-indazole-3-carbonyl]valinate), 5F-MDMB-PINACA (**2**) (5F-ADB; methyl 2-[1-(5-fluoropentyl)-1*H*-indazole-3-carboxamido]-3,3-dimethylbutanoate), 5F-MDMB-PICA (**3**) (methyl 2-[1-(5-fluoropentyl)-1*H*-indole-3-carboxamido]-3,3-dimethylbutanoate), 4F-MDMB-BINACA (**4**) (4F-ADB, 4F-MDMB-BUTINACA; methyl 2-[1-(4-fluorobutyl)-1*H*-indazole-3-carboxamido]-3,3-dimethylbutanoate) and MDMB-4en-PINACA (**5**) (5-CL-ADB-A; methyl 3,3-dimethyl-2-[1-(pent-4-en-1-yl)-1*H*-indazole-3-carboxamido] butanoate) [22,23,24,25].

SCRAs have been detected on the illicit market as pure substances, but are more commonly detected following infusion into herbal materials, papers and e-liquids for smoking and vaping [22,26,27,28]. Valinate and *tert*-leucinate indole and indazole-3-carboxamide SCRAs are chiral compounds, and in all SCRAs studied to date, the (*S*)-enantiomer is significantly more potent than the (*R*)-enantiomer. Chiral profiling data indicate that the SCRAs detected in the majority of samples tested are essentially enantiopure (*S*)-enantiomer however the presence of up to 16% (*R*)-enantiomer has been reported in a small number of cases [1,29,30,31].

An understanding of the physicochemical parameters underpinning the behavior of SCRAs in biological systems, their pharmacokinetics and their effects is essential. For example, the proportion of the SCRA dose that is not plasma protein-bound (the free drug) will be responsible for the drug’s pharmacological effects. This unbound fraction will also be available for metabolism by hepatic and, potentially, extrahepatic enzymes. Whilst a great deal is known about the pharmacokinetics of naturally occurring cannabinoids such as Δ9-tetrahydrocannabinol (THC) [32,33,34,35,36] and its metabolites, less is known about the pharmacokinetics of the valinate and *tert*-leucinate indole and indazole-3-carboxamide SCRAs [37,38,39]. Like SCRAs, the most common routes of administration of THC are smoking and vaping. THC is more bioavailable via inhalation compared to ingestion, and inhalation avoids first-pass metabolism leading to rapid onset of psychoactive effects [34,36]; is highly lipophilic (log P = 6.7 [39]) and has a low distribution into erythrocytes and a high plasma protein affinity (95–99% protein-bound in plasma) [33]. Synthetic cannabinoids are also lipophilic, with theoretical log P values in the literature ranging from 3.02–8.14 [18,19,20,21], although theoretical log P values reported for the valinate and *tert*-leucinate indole-/indazole-3-carboxamides indicate that they are amongst the least lipophilic of the SCRAs.

THC is distributed into the adipose tissue, especially in chronic users, with subsequent slow release from the adipose leading to long detection windows for THC and its metabolites [33,36,40]. Many SCRAs are also highly lipophilic; JWH-210 (4-ethylnaphthalen-1-yl-(1-pentylindol-3-yl)methanone) (log P = 7.5) and RCS-4 (2-(4-methoxyphenyl)-1-(1-pentyl-indol-3-yl)methanone) (log P = 5.6) have shown similar distribution to THC into the adipose tissue of experimentally exposed pigs [39], and long detection windows for other SCRAs in humans have also been reported [41,42,43]. However, lipophilicity is not the only determining factor for uptake by adipocytes [39], as some highly lipophilic drugs are not extensively distributed into adipose tissue [44,45] and structural features are also thought to contribute [46,47].

Castaneto et al. (2015) provided a thorough review of the pharmacokinetics of earlier emerging SCRAs [12]; however, limited/estimated human in vivo pharmacokinetic data are available for currently prevalent SCRAs. The available data from in vivo pharmacokinetic studies for valinate and *tert*-leucinate indole-/indazole-3-carboxamides are based either on long-term abstinence monitoring in casework from users reported to have smoked or vaped the substances and then abstained from their use [43], or on small scale self-administration studies where the drug is ingested rather than smoked or vaped and this does not reflect user behaviour or chronic use [43,48,49,50].

Our understanding of the structure-activity relationships (SARs) and structure-metabolism relationships (SMRs) of the valinate and *tert*-leucinate indole and indazole-3-carboxamide SCRAs is increasing, and phase I metabolite formation is well characterised in vitro [1,2,18,23,31,37,38,51,52,53,54,55]. A valuable systematic study of the SMRs of valinate and *tert*-leucinate indole- and indazole-3-carboxamide SCRAs using pHLM has recently been published [37]. In SCRAs where a valine (dimethyl) methyl ester (AMB-) head group is present, the most abundant metabolite formed is the carboxylic acid metabolite; where a *tert*-leucine (trimethyl) methyl ester (MDMB-) head group is present the carboxylic acid metabolite is formed but is rarely the most abundant metabolite in vitro (MDMB-FUBINACA (**6**) (methyl 2-[1-(5-fluoropentyl)-1*H*-indazole-3-carboxamido]-3,3-dimethylbutanoate) is an exception, probably due to the metabolic stability of the fluorobenzyl moiety). Where a valinamide (dimethyl) (AB-) head group is present, the carboxylic acid metabolite is formed to a greater extent when an indazole core is also present, compared to an indole core, but is never the principal metabolite. Where a *tert*-leucinamide (trimethyl) (ADB-) head group is present the carboxylic acid metabolite is only ever a minor metabolite in vitro, with the most prevalent metabolite determined by the relative lability of the other structural features present [37]. Oxidative defluorination of 5F-MDMB-PINACA (**2**), forming a major metabolite in vitro and in vivo, has been shown to occur in HLM without the presence of NADPH, suggesting the involvement of non-CYP enzymes [54]. The phase I metabolism of 4F-MDMB-BINACA (**4**) has recently been described [55] and was found to involve several cytochrome P450 and human carboxylesterase 1 (CES-1) isoforms, with the carboxylic acid metabolite and formation of a lactone among the most abundant metabolites.

Recently, valinate and *tert*-leucinate indole and indazole-3-carboxamide SCRAs with alkene ‘tail’ groups (MDMB-4en-PINACA (**5**) and AMB-4en-PICA (**7**) (MMB-4en-PICA, MMB-022; methyl 3-methyl-2-[(1-pent-4-enylindole-3-carbonyl)amino] butanoate)) have been detected on the illicit market and in toxicological samples [56,57]. Metabolite identification studies of such SCRAs and other drugs containing an alkene moiety have demonstrated the in vitro formation of a dihydrodiol metabolite (via epoxidation catalysed by CYP isoenzymes, followed by hydration of the epoxide likely facilitated by CYP enzymes and epoxide hydrolase [56,58,59]), and specifically for SCRAs the formation of the carboxylic acid metabolite (through ester hydrolysis), and a carboxylic acid with dihydrodiol metabolite [56]. The carboxylic acid metabolite of MDMB-4en-PINACA (**5**) was the only metabolite detected in blood taken for toxicological analysis from an authentic user, whilst both the carboxylic acid and carboxylic acid with dihydrodiol metabolites were detected in urine. Studies have shown that the carboxylic acids are formed in vitro without the presence of NADPH [54,60,61], and their hepatic formation is thought to be mediated by carboxylesterases, principally CES-1 [55,61,62]. However, interpretation of the detection of carboxylic acid metabolites in plasma in isolation, formed by hydrolysis of the methyl ester and amide moieties, should be treated cautiously. Some SCRAs have been shown to be unstable in human blood when samples have not been frozen prior to analysis [63,64,65], and carboxylesterases are not present in human blood [66,67]. The presence of the carboxylic acid metabolites of these and similar SCRAs in blood could therefore be explained, at least in part, by non-enzymatic hydrolysis or by the action of other plasma esterases prior to, or during, storage, rather than as a result of the recirculation of Phase I metabolites in the body after formation, primarily in the liver [64,67,68].

This study aims to increase understanding of the factors influencing the metabolism and pharmacokinetics of valinate or *tert*-leucinate indole- and indazole-3-carboxamide SCRAs. Their lipophilicity (Log P/Log D_7_._4_) was determined using both in silico methods and experimentally; their short-term stability in plasma has been assessed in vitro under physiological conditions; and their plasma protein binding (PPB) values were determined experimentally for the first time. In vitro intrinsic clearance was calculated following incubation with pooled human liver microsomes (pHLM) and pooled human cryopreserved hepatocytes (pHHeps) for 12 SCRA enantiomer pairs, to investigate the structural and conformational features influencing their interaction with metabolic enzymes and their metabolic clearance rates. In vitro pharmacokinetic data were then used to estimate in vivo human hepatic clearance and hepatic extraction ratios, thus helping to predict their pharmacokinetics in a manner relevant to toxicological casework.

## 2. Results and Discussion

### 2.1. Lipophilicity

Theoretical log P values were calculated for the 12 SCRAs included in this study (Figure 1) using a range of software packages (see Section 3 for details), and the chromatographic hydrophobicity index (CHI) log D at pH 7.4 (Log D_7.4_) was determined experimentally for each (*S*)-enantiomer for comparison. As the tested SCRAs are non-ionizable, log P and log D_7.4_ are equivalent. Although predicted values varied between in silico prediction packages, the ranking of compounds was comparable. AB-FUBINACA (**8**) (*N*-(1-amino-3-methyl-1-oxobutan-2-yl)-1-(4-fluorobenzyl)-1*H*-indazole-3-carboxamide) was the least lipophilic SCRA tested (Log P range = 2.66–3.24) with AMB-CHMICA (**9**) (methyl [1-(cyclohexylmethyl)-1*H*-indole-3-carbonyl]valinate) the most lipophilic (Log P range = 3.84–5.51) (Table 1). The predicted data are similar to previously reported in silico data for valinate and *tert*-leucinate indole and indazole-3-carboxamide SCRAs (2.29–3.81) [19], calculated using ChemBioDraw (Cambridge Soft Corporation, Cambridge, MA, USA).

Experimental log D_7.4_ values (Table 1) ranged from 2.81 (AB-FUBINACA (**8**)) to 4.95 (MDMB-4en-PINACA (**5**)). Theoretical log P values for SCRAs with an indole core were consistently higher than those with an indazole core. However, this was not the case with the experimentally derived data. Predictions were in the closest agreement for 5F-MDMB-PICA (**3**) and AB-FUBINACA (**8**). For all other compounds, experimental values were generally under-predicted by in silico methods. With the exception of MDMB-4en-PINACA (**5**), which was the most lipophilic SCRA studied according to experimental data, all predicted values were within one log unit of experimental values.

It has been reported that it may not be possible to detect parent SCRAs with a log P of 4–5 or greater in urine [19]. The majority of parent valinate and *tert*-leucinate indole and indazole-3-carboxamide SCRAs in this study have log P or log D values at or below this range; they have often been detected in urine, but usually as a very small proportion compared to their metabolites [43,64,69,70,71,72,73]. However, a recent study, involving the oral consumption of the 7-azaindole valinamide-based SCRA, 5F-AB-P7AICA, unusually identified the parent compound as the major component present in urine [50]. Although not included in this study, this SCRA was estimated to have an average theoretical log P value of 2.15 ± 0.46 (Table 1) and so was the least lipophilic molecule measured. This may explain why the parent drug is detected in urine at a higher proportion than its metabolites. In addition, it has been postulated that the presence of an azaindole core structure may decrease the extent to which metabolism occurs at other sites on the structure [50].

### 2.2. Plasma Stability Studies

The instability of valinate and *tert*-leucinate indole and indazole-3-carboxamide SCRAs in unfrozen whole blood and plasma samples has been reported previously [37,38,63], whilst other somewhat contradictory reports have shown long-term stability for some analogs (315 days) at temperatures of up to 20 °C in human plasma [65]. The stability of the majority of the SCRAs described in this study in pooled human plasma (some (*R*)-enantiomers were not tested) over three to five hours under physiological conditions (37 °C, pH 7.4) was assessed prior to carrying out the five-hour long PPB studies. Procaine, the positive control, had a half-life of 6.5 min with no esterase inhibitors present, in line with other reports and the values expected in our laboratory [74,75], and was stable when plasma was pre-incubated with esterase inhibitors.

Over 85% of the parent compound remained following three hours of incubation (Table 2 and Figure 2, with figures for all compounds studied shown in Appendix A).

No structure-stability relationship was observed, there was no difference in the stability between enantiomers, and there was no substantial difference between the incubations with or without inhibitors over the three-hour stability study. The plasma stability of (*S*)-AMB-FUBINACA (**1**) was tested for five hours, the time used in this study for PPB studies, due to its previously reported instability in casework [63]. After five hours, 94 ± 0.1% (*n* = 3) of the compound remained when incubated with esterase inhibitors, and 86 ± 4.0% (*n* = 4) when incubated without esterase inhibitors. This suggests that AMB-FUBINACA may be degraded by plasma esterases over extended time periods, however further study using a longer plasma incubation time is required.

The instability of methyl ester containing SCRAs in plasma has previously been linked to the presence of CES-1 enzymes [76], however, whilst present in the liver and lung, CES-1 is not present in human plasma [66,67]. Human plasma contains other esterases such as butyrylcholinesterase (BchE), paraoxonase (PON-1), albumin esterase (a ‘pseudo-esterase’) and acetylcholinesterase (AchE) [66,67,68] and therefore it is these enzymes that may be involved in the degradation of SCRAs with methyl ester moieties, although non-enzymatic hydrolysis may also be involved [64].

### 2.3. Plasma Protein Binding (PPB)

The experimentally derived PPB values for the majority of SCRAs in this study are shown in Table 3 (some (*R*)-enantiomers were not tested due to reference standard availability). All SCRAs were highly protein bound, with PPB ranging from 88.9–99.5%. For the positive controls, warfarin and nicardipine, binding values were 98.4% and 97.7%, respectively, in agreement with literature data [77,78]. The observed binding of the SCRAs reported in this study were in line with plasma protein binding data previously reported for THC [33] and is related to the lipophilicity of the SCRAs tested [13,79]. Thermal stability controls, which consisted of spiked plasma stored at 4 °C and 37 °C for the duration of the experiment, showed that most compounds remained stable (total peak area in dialysed sample >95% of the peak area in the non-dialysed sample). SCRAs for which recovery (total peak area in dialysed samples) was ≤95% that of 4 °C samples were all valine methyl ester (AMB-) compounds ((*S*)-AMB-FUBINACA (**1**) recovery = 77 ± 4.4%, *n* = 3; (*R*)-AMB-4en-PICA (**7**) recovery = 95 ± 2.0%, *n* = 3). This agrees with available plasma stability data for the 3- and 5-h incubations reported here and with published long-term stability studies [63,65]. These findings all support the view that SCRAs containing a valine methyl ester head group are less stable in plasma than those containing a *tert*-leucine methyl ester or *tert*-leucinamide/valinamide head group.

As the equilibrium dialysis method employed to determine PPB involves the measurement of the ratio of bound to unbound analyte at the end of the equilibration period, the instability observed for the valine methyl ester SCRAs will not affect the calculated binding value [80].

### 2.4. In Vitro Intrinsic Clearance

The in vitro metabolic stability of the tested (*S*)- and (*R*)-SCRA enantiomers in pHLM and pHHeps incubations are summarised in Table 4 and Table 5, respectively. Each batch of pHLM and pHHeps used in the study comprised pooled biological material from 50 donors, 25 male and 25 female, of varying ages and health status. Details of the donors to each batch are provided in the certificate of analysis associated with each batch (see Section 3.1 for further details of donors). It is recognized that there will be inter- and intra-individual variation in SCRA metabolism, due to age, sex, health status and polymorphisms [81]. The use of large donor pools for pHLM and pHHeps batches (*n* = 50) reduces inter-batch variation and provides information on the fundamental structure metabolism relationships of SCRA analogues within and between structural classes, and intrinsic clearance rates obtained relate to the enzyme expression of an ‘average individual’. This study does *not* set out to study differences in the ability of individuals, or phenotypes to metabolise SCRAs. Positive controls included with all test batches showed acceptable clearance; verapamil CL_int_ in pHLM ranged from 153–360 mL min^−1^ kg^−1^, while CL_int_ in cryopreserved pHHeps ranged from 117–179 mL min^−1^ kg^−1^, within the expected ranges for our laboratory and in line with literature reports [82,83,84,85,86]. Further positive controls for pHHep incubations, 7-ethoxycoumarin and 7-hydroxycoumarin, provided intrinsic clearance rates ranging from 55–305 mL min^−1^ kg^−1^ and 46–238 min^−1^ kg^−1^, respectively. The ranges observed cover normal batch variation for the metabolism of these compounds. Hepatocyte viability ranged from 90–94%. All compounds were initially tested using a single pHHeps lot (HUE50-N, see Section 3.1 for further information on pool donors), allowing determination of fundamental differences in metabolic stability between compounds. A subset of compounds was further tested at a later date using pHHeps lot HUE50-P. This gave an indication of potential between-batch variation, evident only for (*R*)-5F-MDMB-PINACA (**2**). Intrinsic clearance and predicted in vivo data calculated using alternative microsomal scaling factors and an alternative hepatocyte cell density reported in the literature are provided for microsome and hepatocyte incubations in Appendix A, respectively.

#### 2.4.1. Comparison of SCRA (*S*)-Enantiomer Intrinsic Clearance Rates and Half-Lives

The (*S*)-enantiomers of valinate and *tert*-leucinate indazole- and indole-3-carboxamide SCRAs are known to be much more prevalent and more potent than the (*R*)-enantiomers and are often more efficacious [1,31]. For the (*S*)-enantiomers tested in this study, the order of the in vitro intrinsic clearance rate was AMB > MDMB > AB-compounds in both pHLM and pHHeps (Table 4 and Table 5; Figure 3 and Figure 4). Data showing the clearance of each enantiomer pair are provided in Appendix A for pHLM and pHHeps, respectively). The in vitro half-lives of SCRAs in pHLM incubations ranged from 118 ± 28 min for (*S*)-AB-FUBINACA (**8**) to 0.60 ± 0.02 min for (*S*)-AMB-FUBINACA (**1**). In pHHeps, half-lives ranged from 76 ± 24 min for (*S*)-AB-FUBINACA (**8**) and 2.50 ± 0.55 min for (*S*)-AMB-FUBINACA (**1**). In vitro pHLM intrinsic clearance for some compounds (e.g., (*S*)-AMB-FUBINACA (**1**) and (*S*)-5F-AMB-PINACA (**10**) (methyl [1-(5-fluoropentyl)-1*H*-indazole-3-carbonyl]valinate)) was so fast that it was challenging to accurately calculate intrinsic clearance rates and half-lives (see Appendix A). The presence of a valinamide head group (AB-compounds) increased drug half-life, slowing clearance compared to SCRAs with a more labile methyl ester moiety (AMB- and MDMB-compounds). However, AB-CHMINACA (**11**) (*N*-[1-amino-3-methyl-1-oxobutan-2-yl]-1-(cyclohexylmethyl)-1*H*-indazole-3-carboxamide) clearance in hepatocytes was comparable to that of the slowest cleared *tert*-leucine methyl ester (MDMB-) compounds, possibly due to the relative lability of the cyclohexyl tail group to enzymatic attack [87]. *tert*-Leucine methyl ester (MDMB-compounds), having a trimethyl moiety in the head group, are metabolised at a slower rate than those with a dimethyl moiety in the head group (AMB-, valine methyl ester compounds). This could be due to steric hindrance caused by the extra methyl group adjacent to the methyl ester, one of the principal sites of biotransformation.

The differences in clearance rates between SCRA compounds which differ only by their head group (AB-, AMB- and MDMB-compounds) and which had either a fluorobenzyl tail group (FUBINACAs) or a fluoropentyl tail group (PINACAs) is illustrated in Figure 4.

Although no *tert*-leucinamide SCRAs (ADB-type SCRAs) were investigated in this study, it might logically be concluded that their intrinsic clearance is likely to be similar or even slower than the valinamide SCRAs (AB-type SCRAs), however this requires further experimental confirmation. The in vitro clearance of SCRAs with indazole cores (“INACA” compounds) is faster in both pHLM and pHHeps than their equivalent indole analogues (“ICA” compounds), e.g., 5F-MDMB-PINACA (**2**) vs. 5F-MDMB-PICA (**3**) and MDMB-4en-PINACA (**5**) vs. MDMB-4en-PICA (**12**) (methyl 3,3-dimethyl-2-[1-(pent-4-en-1-yl)-1*H*-indole-3-carboxamido]butanoate) (Table 4 and Table 5).

Overall, these findings corroborate data from a number of previous studies: Franz et al. (2019) [37] demonstrated that indoles were significantly less metabolically reactive than their indazole analogues during in vitro pHLM studies; Hess et al. (2017) [65] demonstrated that long term stability in non-frozen plasma (4 °C and 20 °C) was of the order AB-FUBINACA = ADB-FUBINACA (*N*-(1-amino-3,3-dimethyl-1-oxobutan-2-yl)-1-(4-fluorobenzyl)-1*H*-indole-3-carboxamide) > MDMB-FUBINACA > AMB-FUBINACA; Krotulski et al. (2020) [63] showed that the order of stability in non-frozen whole blood was ADB-FUBINACA > 5F-MDMB-PICA > 5F-MDMB-PINACA > AMB-FUBINACA.

The reactivity/lability of the methyl ester and amide moieties within the valinate and *tert*-leucinate indole and indazole-3-carboxamide SCRAs appears to be the determining step for their in vitro clearance rates, the type of metabolites formed in vitro and in vivo and their stability ex vivo in whole blood and plasma. It is therefore logical that in both this study and in other studies, AMB-FUBINACA (**1**) has been found to be among the most metabolically unstable compounds studied. The reasons being that the methyl ester moiety is fundamentally unstable; there is little steric hindrance to slow down enzymatic hydrolysis of the methyl ester by esterases; the molecule is relatively more reactive due to its indazole core compared to indole compounds, and other sites of metabolism on the molecule are limited due to the presence of the stable fluorobenzyl moiety.

#### 2.4.2. The Influence of Chirality on Intrinsic Clearance Rates and Half-Lives

As enantioselectivity is common in both metabolism and pharmacology [88], comparison of the intrinsic clearance and half-lives between enantiomer pairs can provide some further fundamental information on the influence of structural and conformational features on the molecular interactions between SCRA substrates and metabolic enzymes.

There were some notable differences in the half-lives and intrinsic clearance derived from the incubation of (*R*)- and (*S*)- enantiomers with pHLM (Table 4, Appendix A). For most compounds, the (*S*)-enantiomer was consistently cleared at a faster rate than the (*R*)-enantiomer, particularly for AMB-FUBINACA (**1**) and 5F-AMB-PINACA (**10**).

For those compounds with an alkene tail (4en compounds), there was either no difference between enantiomers (MDMB-4en-PINACA (**5**)) or the (*R*)-enantiomer was cleared faster (AMB-4en-PICA (**7**) and MDMB-4en-PICA (**12**)). When SCRA enantiomer pairs were incubated in pHHeps (Table 5, Appendix A), there was negligible difference between the derived half-lives and intrinsic clearance rates of (*S*)- and (*R*)-enantiomers for AB-compounds. For AMB-compounds there was a marginal difference between the intrinsic clearance and half-lives of the enantiomer pairs with (*S*)-enantiomers cleared at a faster rate. *tert*-Leucine methyl ester SCRAs without an alkene tail group (5F-MDMB-PINACA (**2**), 5F-MDMB-PICA (**3**), 4F-MDMB-BINACA (**4**) and MDMB-FUBINACA (**6**)), showed the greatest difference in intrinsic clearance rates between the (*S*)- and (*R*)-enantiomers, with the (*S*)-enantiomers consistently cleared at a faster rate. Of the *tert*-leucine methyl ester SCRAs with an alkene tail group (-4en- compounds), the (*S*)-enantiomer of MDMB-4en-PINACA (**5**) was cleared faster than the (*R*)-enantiomer but the intrinsic clearance of the enantiomer pairs of AMB-4en-PICA (**7**) and MDMB-4en-PICA (**12**) was similar.

In this study, the majority of pHHep incubations were carried out using hepatocytes of a single lot number, however 5F-MDMB-PINACA (**2**) was tested using two different lots of pHHep cells. In initial pHHep incubations (lot HUE50-N), incubation of (*S*)-5F-MDMB-PINACA and (*R*)-5F-MDMB-PINACA (**2**) gave half-lives of 13.3 ± 3.4 min (*n* = 8) and 22.9 ± 1.8 min (*n* = 6) respectively, while in pHHep lot HUE50-P, half-lives were 13.2 ± 3.4 min (*n* = 4) and 6.5 ± 0.13 min (*n* = 4), respectively. The increased clearance of (*R*)-5F-MDMB-PINACA (**2**) in the latter incubation suggests a difference in expression of enzyme(s) in this lot of cells, but no such differences were observed for the other compounds tested (MDMB-4en-PINACA (**5**), MDMB-4en-PICA (**12**) and AMB-4en-PICA (**7**)) using the two different pHHep lots.

#### 2.4.3. Comparison of Intrinsic Clearance Calculated from pHLM and pHHeps

For all compounds, with the exceptions of (*R*)-5F-MDMB-PICA (**3**) and (*R*)-MDMB-4en-PICA (**12**), intrinsic clearance scaled to whole-liver dimensions for pHHeps was faster than or comparable to that of pHLM. For all MDMB- compounds, and for most (*S*)-AMB-compounds differences in CL_int_ were within 2.5-fold. Those containing an amide (AB-compounds) rather than a methyl ester group, and thus forming an amide hydrolysis product as a major metabolite [69,87], showed the greatest differences in intrinsic clearance between pHLM and pHHeps but were cleared the slowest overall. There are multiple factors which can cause intrinsic clearance by HHeps to differ from HLM [89]. Differences may reflect the relative efficiency of transport of SCRAs and their enantiomers through cell membranes in HHeps and the relative abundance of particular enzymes in HLM and HHeps, targeting different sites of metabolism, particularly carboxylesterases. A dominance of non-CYP enzymes in metabolic clearance can result in CL_int_ being faster in HHeps. The degree of Phase II glucuronidation thought to be produced from these drugs varies, but glucuronidation here only occurs after some form of Phase I biotransformation and is generally minimal compared to other pathways [56,64,69,70,71,90]; these studies however did not utilise metabolite reference standards or beta-oxidation for confirmation. Glucuronidation is therefore unlikely to cause the differences in clearance rates observed. Carboxylesterase enzymes are non-CYP enzymes, but are present in microsomes and do not require cofactor [91]. CES-1 is one of the most abundant enzymes in human hepatocytes, present in the endoplasmic reticulum and to a lesser extent in the cytosol [67,88,91]. Non-specific binding of drugs to microsomes and hepatocytes results in a lower unbound fraction available to interact with enzymes; differences in binding between HLM and HHep incubations may therefore also cause clearance differences [89]. Non-specific binding may be estimated from log P data [92,93]. Theoretical fraction unbound for microsomes (fu_mic_) and hepatocytes (fu_hep_) was calculated using experimental pHLM and pHHep concentrations and incubation volumes, and experimental log D_7.4_ values, as described by Kilford et al. (2008) [92] with hepatocyte cell volume as 3.9 µL/million cells [94]. In this study, fu_mic_ ranged from 0.18–0.82 (*n* = 12, median = 0.42), while fu_hep_ ranged from 0.47–0.95 (*n* = 12, median = 0.75). These results suggest in vivo hepatocyte binding is mostly limited (except, perhaps for MDMB-4en-PINACA (**5**) (estimated fu_hep_ 0.47)), while microsomal binding is greater. However, differences in predicted binding did not necessarily correlate with differences in clearance, so this is unlikely to be a significant factor.

### 2.5. Prediction of Human In Vivo Clearance

When PPB was considered in the prediction of in vivo hepatic clearance (see Table 3), clearance rates were, as would be expected, much slower than in vitro values (Table 4 and Table 5). Hepatic clearance predicted from pHLM incubations ranged from 0.34 mL min^−1^ kg^−1^ ((*S*)-AB-FUBINACA (**8**)) to 17.79 mL min^−1^ kg^−1^ ((*S*)-5F-AMB-PINACA (**10**)). Hepatic clearance predicted from pHHep incubations ranged from 1.39 mL min^−1^ kg^−1^ ((*S*)-MDMB-FUBINACA (**6**)) to 18.25 mL min^−1^ kg^−1^ ((*S*)-5F-AMB-PINACA (**10**)). Predicted in vivo hepatic clearance from HLM in vitro data were calculated to allow comparison with available literature values. They should be viewed cautiously for SCRAs with a terminal ester or amide moiety due to likely differences in CES-1 enzyme expression between pHLM and pHHeps. However, in this study, no marked differences in predicted in vivo clearance rates between pHLM and pHHeps were observed. SCRA clearance studies have previously shown that true in vivo clearance is often slower than is predicted from in vitro data [15,95]. This is likely due to the lipophilicity and high degree of plasma protein binding of these compounds; for most SCRAs, PPB has not previously been characterised and so was not previously incorporated into the prediction of human in vivo rates. Hepatic clearance in this study has been predicted according to the ‘well-stirred’ model, where *fu* may be based on either blood or plasma binding data [79,96]. Here, plasma binding data were used and so it is assumed that the blood/plasma ratio of the test compounds is 1. Differences in binding in plasma versus whole blood could therefore influence the results. Hepatic extraction ratios (Table 4 and Table 5) showed large differences between the compounds tested. The ratios ranged from 0.02 ((*S*)-AB-FUBINACA (**8**)) to 0.85 ((*S*)-5F-AMB-PINACA (**10**)) based on pHLM incubations, and from 0.07 ((*S*)-MDMB-FUBINACA (**6**)) to 0.87 ((*S*)-5F-AMB-PINACA (**10**)) based on pHHep incubations. The fraction of drug predicted to be cleared with each pass of the liver therefore varies greatly between compounds. Although (*S*)-enantiomers tended to have faster intrinsic clearance rates in pHHeps, they also tended to be slightly more protein bound than (*R*)-enantiomers; thus, predicted in vivo clearance rates and hepatic extraction ratios were not substantially different between enantiomers.

For a small number of the SCRAs tested in this study, comparative intrinsic clearance data are available in the literature (Table 6), and data reported here was in overall agreement with the reported values. While studies of intrinsic clearance in pHHeps are limited, more studies have reported microsomal rates. Although not reported, it is assumed that the (*S*)-enantiomers have been tested in the literature studies. Castaneto et al. reported AB-FUBINACA (**8**) CL_int micr_ as 0.011 mL min^−1^ mg^−1^, with intrinsic clearance of 10.5 mL min^−1^ kg^−1^ [69]. Presley et al. found 5F-MDMB-PINACA (**2**) CL_int micr_ to be 0.271 mL min^−1^ mg^−1^, with an intrinsic clearance rate of 256.2 mL min^−1^ kg^−1^ [54]. These findings are in line with data reported here. However, as protein binding studies were unavailable at the time, CL_H_ and E_H_ predictions by Castaneto et al. and Presley et al. [54,69] did not account for binding and so rates for this differ; based on pHLM incubations, CL_H_ and E_H_ predictions for AB-FUBINACA (**8**) [69] were 20-fold and 17-fold lower respectively when PPB was accounted for, while those for 5F-MDMB-PINACA (**2**) [54] were 4-fold lower in this study. Intrinsic clearance of 5F-AMB-PINACA (**10**) and AMB-4en-PICA (**7**) have also been reported previously, and both were similar to the data reported in this study [57,73]. Davidsen et al. (2010) [13] reported increasing metabolic stability of JWH-type SCRAs in HHeps with increasing size of halogen atom incorporated into the structure, with I-JWH-122 ((1-(5-iodopentyl)-1*H*-indol-3-yl)(4-methylnaphthalen-1-yl)methanone) being the most metabolically stable. The effect of incorporation of fluorine onto the pentyl tail group is unclear; for JWH-122 ((4-methyl-1-naphthyl)-(1-pentylindol-3-yl)methanone) (HHeps) and AMB-PINACA (methyl (1-pentyl-1*H*-indazole-3-carbonyl)valinate) (HLM), 5-fluorination makes little difference to intrinsic clearance [13,73].

The 5-fluorination of Cumyl-PICA (1-pentyl-*N*-(2-phenylpropan-2-yl)-1*H*-indole-3-carboxamide) (HLM) results in faster clearance [95], while 5-fluorination of AB-PINACA (HLM) results in slower clearance [16]. As noted previously, the incorporation of fluorobenzene (e.g., FUB- or -FUBINACA compounds) tends to stabilise compounds, reducing clearance rates [16,69,97].

In vivo hepatic clearance is related to both CL_int_ and fu [79]; for many drugs, increasing log P will increase CL_int_ but reduce fu. Thus, the effect of log P on whole-body clearance can be unpredictable. However, increasing log P can increase the volume of distribution of the unbound fraction [79]. Although the majority of the SCRAs studied here are rapidly metabolised in vitro, high levels of protein binding in vivo will extend the time they will circulate in the body following consumption. This potentially facilitates storage in lipid-rich tissues such as adipose tissue, particularly in chronic users. Detection windows of some compounds have been reported to be exceptionally long following drug cessation [43,98,99]. Franz et al. reported detection in urine of a metabolite common to AB-FUBINACA (**8**) and AMB-FUBINACA (**1**) two years after reported cessation of consumption, but did not determine which drug was responsible [43]. Hasegawa et al. (2015) reported detection of AB-CHMINACA (**11**) and 5F-AMB-PINACA (**10**) in a fatal poisoning case, where AB-CHMINACA (**11**) could be quantified in all solid tissues tested, while 5F-AMB-PINACA (**10**) could only be quantified in adipose tissue [98]. These data suggest extensive redistribution of synthetic cannabinoids in vivo. Experimental data presented here show hepatic clearance of AB-type SCRAs to be slower than that of MDMB- and AMB- compounds, with lower hepatic extraction ratios. However, the SCRAs are not metabolically stable enough to explain the recently reported long detection windows. Therefore, the presence of such compounds and their metabolites in body fluids after a considerable time is more likely to be evidence of tissue accumulation, subsequent leaching back into the circulatory system and metabolism of the unbound fraction. JWH- SCRAs have log P values which are similar to or greater than those of indole- and indazole-3-carboxamide SCRAs (and therefore likely similar or greater PPB), which may in part explain their extended detection windows in vivo [100,101], though specific structural features are likely also involved [39].

### 2.6. Study Limitations

A limitation to this study, and indeed other research studies related to the metabolism of constantly emerging SCRAs and other NPS using pHLM and pHHeps, is that it does not consider the expected variation in metabolic capacity between individual users. The aim of the study is, however, to provide comparative information for the metabolism of structurally related SCRAs in an ‘average’ human using pooled donor in vitro platforms, to allow the deconvolution of the key structural features of the SCRAs that affect metabolic stability in isolation. The study provides a more accurate estimate of in vivo human hepatic clearance than previously reported by taking into account the effect of plasma protein binding. This is the first time such information has been incorporated into in vivo hepatic clearance calculations for the SCRA structural classes studied. The study also does not consider other potential sites of metabolism likely to be important in SCRAs which are smoked or vaped, namely the lungs; no metabolic studies on SCRAs using lung microsomes have yet been reported in the literature. Metabolism by the lungs could occur to some degree as they contain numerous metabolising enzymes including, but not limited to, carboxylesterases such as CES-1 (but at a lower concentration than the liver) [102], likely to be the main enzyme responsible for methyl ester and amide hydrolysis of these SCRAs. The lungs also contain CYP450 enzymes as well as uridine 5′-diphospho-glucuronosyltransferase (UGT) enzymes (but with limited expression compared to other tissues) [103]. While Phase I metabolism in human lung parenchymal cells has shown less than 10% the activity of that of cryopreserved human hepatocytes, levels of ester hydrolysis, of great importance to the valinate- and *tert*-leucinate SCRAs included in this study, have been shown to be similar [104]. Through inhalation of SCRAs, the lungs will be exposed to high drug concentrations. It is unclear to what extent metabolism of valinate and *tert*-leucinate indole and indazole-3-carboxamide SCRAs in the lungs may affect the extent of SCRA biotransformation, and thus the bioavailability of the parent compound, before reaching systemic circulation.

**Table 6 molecules-26-01396-t006:** Literature values for intrinsic clearance (CL_int_) rates and half-lives (t_1/2_) of SCRAs in human liver microsomes (HLM) and human hepatocytes (HHeps). Compounds marked with an asterisk (*) display pHHep data; all other data is from pHLM incubations.

Compound	HLM/HHeps *			Reference
	CL_int micr_ (mL min^−1^ mg^−1^)	CL_int_ (mL min^−1^ kg^−1^)	t_1/2_ (min)	
JWH-122 *	-	1350	0.95	[13]
MAM-2201 *	-	1408	0.88	
Cl-JWH-122 *	-	502	2.46	
Br-JWH-122 *	-	497	2.49	
I-JWH-122 *	-	235	5.26	
AM1220 *	-	169	3.7	[105]
THJ-018	0.036	34.2	19.2	[14]
THJ-2201	0.064	60.8	10.8	
NNEI	0.300	350	2.29	[15]
MN-18	0.410	469	1.71	
NM-2201	0.088	81.6	8.0	[106]
BIM-2201	0.142	134.1	4.9	[107]
Cumyl-PICA	0.12	-	5.92	[95]
5F-Cumyl-PICA	0.39	-	1.77	
STS-135	0.222	209	3.1	[108]
AMB-4en-PICA (**7**)	-	291	2.1	[57]
AB-PINACA	0.037	35	18.7	[16]
5F-AB-PINACA	0.019	18	35.9	
AMB-PINACA		-	1.1	[73]
5F-AMB-PINACA (**10**)	0.67	-	1.0	
5F-MDMB-PINACA (**2**)	0.271	256.2	3.1	[54]
5F-APP-PICA	0.046	-	15.1	[109]
5F-APP-PINACA	0.202	-	3.4	
APP-CHMINACA	0.133	-	5.2	
AB-FUBINACA (**8**)	0.011	10.5	62.6	[69]
ADB-FUBINACA	0.018	16.5	39.7	[17]
FDU-PB-22	0.056	52.7	12.4	[97]
FUB-PB-22	0.060	57.1	11.5	

## 3. Materials and Methods

### 3.1. Chemicals and Reagents

The in-house enantiospecific synthesis of the (*R*)- and (*S*)-enantiomers of AMB-FUBINACA (**1**) (>98% purity), 5F-MDMB-PINACA (5F-ADB) (**2**) (>99.6% purity), 5F-MDMB-PICA (**3**) (>99% purity), 4F-MDMB-BINACA (**4**) (>99.7% purity), MDMB-4en-PINACA (**5**) (>98.6% purity), MDMB-FUBINACA (**6**) (>99.9% purity), AMB-4en-PICA (**7**) (>99.7% purity), AB-FUBINACA (**8**) (>99% purity), AMB-CHMICA (**9**) (MMB-CHMICA) (>99.6% purity), 5F-AMB-PINACA (**10**) (5F-AMB) (>99.9% purity), AB-CHMINACA (**11**) (>99% purity) and MDMB-4en-PICA (**12**) (>99.7% purity) has been described previously [1,31].

Donepezil, verapamil, 7-ethoxycoumarin (7-EC), bis(4-nitrophenyl) phosphate (BNPP), phenyl methyl sulphonyl fluoride (PMSF), procaine, warfarin, nicardipine, nicotinamide adenine dinucleotide phosphate (NADP), reduced NADP (NADPH), glucose-6-phosphate dehydrogenase (from baker’s yeast *S. cerevisiae*), d-glucose-6-phosphate sodium salt, sodium bicarbonate, potassium phosphate buffers (monobasic and dibasic), formic acid (99%) and dimethylsulfoxide (DMSO) were purchased from Sigma-Aldrich (Gillingham, UK). 7-hydroxycoumarin (7-HC) was purchased from Chem Service (West Chester, PA, USA). Analytical grade acetonitrile (ACN) and methanol (MeOH) were purchased from Sigma-Aldrich or Greyhound Chromatography (Birkenhead, UK).

Ultra-high purity water (18 MΩ cm^−1^) was obtained using a Milli-Q water purification system (Merck, Livingston, UK). Human plasma (pooled; Na EDTA anticoagulant; lot numbers 27079, IR07-081) (Innovative Research (Novi, MI, USA)) was purchased from Patricell (Nottingham, UK)). Pooled human liver microsomes (pHLM, donor pool 50, lots PL050E-A, PL050E-B with epidemiological information available for all donors [110,111]), pooled cryopreserved human hepatocytes (pHHeps, donor pool 50, lots HUE50-N, HUE50-P with epidemiological information available for all donors [112,113]), Williams Media E (WME) (without glutamine or phenol red), cell maintenance supplement pack (with Cocktail B/Dexamethasone), trypan blue and cryopreserved hepatocyte recovery medium were purchased from Gibco (ThermoFisher Scientific, Waltham, MA, USA). pHLM preparations were incubated in a Stuart S160 benchtop incubator with shaking on a Stuart SSM1 mini orbital shaker (Cole-Parmer, St. Neots, UK). pHHep preparations were incubated in a Thermo Scientific HERAcell Vios 160i CO_2_ incubator, with shaking on a Thermo Scientific CO_2_ resistant orbital shaker (Fisher Scientific, Loughborough, UK). Individual reference standard stock solutions were prepared at 1–10 mM in DMSO depending on purpose and stored at −20 °C until use, unless used immediately. Working solutions were prepared on the day of use.

### 3.2. Instrumentation

Chromatographic hydrophobicity index (CHI) log D samples (see below) were analysed using high-performance liquid chromatography (HPLC) with a Shimadzu Nexera X2 HPLC coupled to a Shimadzu SPD-30 MA photodiode array detector (PDA) (Shimadzu, Kyoto, Japan). Data analysis was performed using Shimadzu LabSolutions 5.91 software. Samples (2 µL) were injected onto an Acquity BEH C_18_ column (2.1 × 50 mm^2^, 1.7 µm) (Waters Corp., Milford, MA, USA) at a flow rate of 0.6 mL min^−1^ at 25 °C. The sample chamber was maintained at 25 °C. Gradient elution was performed, with mobile phase (MP) consisting of ammonium acetate in water (10 mM, pH 7.4) (MP A) and acetonitrile (MP B). The following gradient was used: 0–4.0 min from 5% B to 95% B, 4.0–4.5 min hold 95% B, 4.5–4.51 min to 5% B, 4.51–5.51 min hold 5% B. Detection was achieved by scanning over a range of 190 to 500 nm wavelength, with 254 nm used for identification and peak area normalisation.

In vitro incubation samples (see below) were analysed using ultra-performance liquid chromatography-tandem mass spectrometry (UPLC-MS/MS) with a Waters Acquity UPLC coupled to a Waters Xevo TQ-S micro tandem mass spectrometer (Waters Corp., Milford, MA, USA). Data analysis was performed using Waters Masslynx 4.1 software (Waters Corp.). Samples (1 µL) were injected onto an Acquity UPLC BEH C_18_ column (2.1 × 50 mm^2^, 1.7 µm) (Waters Corp.) at a flow rate of 0.6 mL min^−1^ at 45 °C. The sample chamber was maintained at 4 °C. Gradient elution was performed, with mobile phases (MP) consisting of water with formic acid (0.01% *v*/*v*, MP A) and methanol with formic acid (0.01% *v*/*v*, MP B). The following gradient was used: 0–0.3 min 5% B, 0.3–1.3 min to 95% B, 1.3–1.79 min hold 95% B, 1.79–1.8 min to 5% B. MS acquisition was achieved with positive electrospray ionisation (ESI) and the MS operating in MRM mode, with MRM transitions provided in Appendix A. The following MS parameters were used: source temperature 100 °C, desolvation gas 500 °C at a flow rate of 1000 L h^−1^, cone gas flow rate 50 L h^−1^, collision gas (nitrogen) at 0.3 mL min^−1^, cone voltage was 44 V and capillary voltage was 3.40 kV.

### 3.3. In Silico Log P Prediction

Log P values were predicted for all SCRAs tested in this study using multiple software packages, with the full dataset provided in Appendix A. Software packages used were SwissADME (Swiss Institute of Bioinformatics, Lausanne, Switzerland; online and free to access), Gastroplus (SimulationPlus, Lancaster, CA, USA; MedChem Designer version 4.5), MoKa (Molecular Discovery, Borehamwood, UK; version 3.0), Canvas (Schrödinger, LLC, New York, NY, USA; version 3.6) and XlogP (Institute of Physical Chemistry, Peking University [114]).

### 3.4. Experimental Log D_7.4_

Log D (pH 7.4) was determined by chromatographic hydrophobicity index (CHI) measurements for the (*S*)-enantiomer of each SCRA involved in this study. A calibration mix of ten compounds at 10 µg mL^−1^ in 1:1 (*v*/*v*) water:ACN was used (theophylline, phenyltetrazole, benzimidazole, colchicine, phenyltheophilline, acetophenone, indole, propiophenone, butyrophenone and valerophenone (all Sigma-Aldrich)). Test compounds were prepared at 0.25 mM in 1:1 (*v*/*v*) water:ACN. Samples (*n* = 2 for calibration standards; *n* = 1 for test compounds) were analysed using UPLC-PDA as described above. Retention times of calibration compounds are used to calculate retention factor (k), which is plotted against literature CHI values for each compound. Linear regression was performed to obtain a line of best fit, from which the CHI and subsequently CHI log D of calibration and test compounds are calculated. Calculations used and calibration data from CHI log D studies are provided in Appendix A.

### 3.5. Plasma Stability Studies

The pooled plasma used for both plasma stability and plasma binding studies originated from 50 donors, collected as whole blood from donors in the United States at an FDA-approved collection center, processed into plasma by centrifugation and immediately frozen (Innovative Research; Novi, MI, USA) and purchased from Patricell (Nottingham, UK). Plasma was tested by the manufacturers for a range of FDA-required viral markers. Pooled human plasma, was centrifuged (3750 rpm, 10 min, 22 °C) and the supernatant buffered to pH 7.4 by diluting to 70:30 plasma to potassium phosphate buffer (50 mM, pH 7.4). Stock solutions were prepared in DMSO. For inhibitor incubations, esterase inhibitors BNPP and PMSF were added to plasma mixture to give final concentrations of 500 µM for each. For incubations without inhibitor an equivalent volume of DMSO was included. Plasma mixture was pre-incubated at 37 °C for 5 min. Test compounds (5 µM) and procaine (positive control, 50 µM) were added to pre-warmed plasma mixture containing inhibitors. Incubation volume was 1 mL. Samples were mixed and 80 µL was immediately sampled and quenched into 200 µL ACN spiked with donepezil (internal standard (IS), 50 ng mL^−1^). Plates were incubated at 37 °C with shaking (100 rpm) and additional aliquots were sampled at 10, 30, 60, 120 and 180 min. (*S*)-AMB-FUBINACA (**1**) was further sampled at 240 and 300 min as instability in plasma has been reported previously [63]. Samples were centrifuged (3750 rpm, 10 min, 22 °C) and the supernatant (150 µL) diluted with 50 µL deionised water before analysis by UPLC-MS/MS. Peak area ratio of analyte to IS was used to determine half-lives of test compounds in plasma, both with and without the presence of esterase inhibitors.

### 3.6. Plasma Protein Binding

Plasma protein binding was assessed by equilibrium dialysis in a 96-well dialysis block (HTDialysis, Ledyard, CT, USA). Dialysis membranes (12–14 kDa) were soaked in deionised water for at least 60 min. Membranes were soaked overnight at 4 °C in ethanol:water 20:80% *v/v* and then rinsed with deionised water immediately prior to use. Pooled human plasma (Innovative Research, Novi, MI, USA) was centrifuged (3750 rpm, 10 min, 22 °C), spiked with analyte at 10 µg mL^−1^ and allowed to equilibrate at room temperature (22 °C) for 20 min. Nicardipine and warfarin were included as positive controls. Spiked plasma (150 µL) was dialysed against isotonic phosphate buffer (pH 7.4) (150 µL) for 5 h at 37 °C with shaking at 100 rpm. Thermal degradation controls for each test compound were kept at 37 °C and 4 °C. Blank control plasma and buffer were included in the incubation. Following incubation, dialysed plasma samples and thermal degradation controls (50 µL) were added to drug-free buffer (50 µL), and dialysed buffer samples (50 µL) were added to drug-free plasma (50 µL). ACN (200 µL) containing donepezil (IS, 50 ng mL^−1^) was added to each sample and samples were centrifuged (3750 rpm, 10 min, 22 °C). Supernatant (150 µL) was added to 50 µL deionised water and samples were analysed by UPLC-MS/MS as described above. Dialysed buffer from each well was tested for protein contamination using BCA protein assay reagent to determine whether membranes had been compromised. Percentage drug bound and fraction unbound were calculated as follows [115]:(1)$$\%\;\mathrm{bound}=\frac{\left(\mathrm{Pl}-\mathrm{Bu}\right)}{\mathrm{Pl}}\;\times 100$$
where Pl = Analyte/IS ratio determined in plasma side

Bu = Analyte/IS ratio determined in buffer side
(2)$$\mathrm{Fraction}\;\mathrm{unbound},\;\mathrm{fu}=\frac{\left(100-\%\;\mathrm{bound}\right)}{100}$$

### 3.7. Metabolic Stability—Pooled Human Liver Microsome Incubations

Test compounds and verapamil (positive control) were incubated at 0.5 µM in pHLM (0.5 mg microsomal protein mL^−1^) incubations. Total incubation volume was 500 µL. Procedure: Test compound working solutions (50 µM in potassium phosphate buffer (pH 7.4)) were prepared (<1% DMSO). pHLM were thawed and diluted in potassium phosphate buffer (pH 7.4). Drug working solution (5 µL) in phosphate buffer (50 µM, final concentration 0.5 µM) and 445 µL microsome suspension (final concentration 0.5 mg mL^−1^) were pre-incubated at 37 °C for 5 min, and 50 µL 8 mg mL^−1^ NADPH in phosphate buffer (final concentration 0.8 mg mL^−1^) was used to initiate the reaction. Plates were incubated at 37 °C with shaking (100 rpm) and samples (50 µL) collected at 0, 3, 6, 9, 15, 30 min. Reactions were quenched in 200 µL ACN spiked with donepezil (IS, 50 ng mL^−1^). Samples were diluted with 50 µL deionised water and sealed and centrifuged (3750 rpm, 10 min, 22 °C) to sediment any precipitated proteins. Samples were analysed by UPLC-MS/MS in a 96 deep-well 2 mL plate. Each SCRA was incubated at least in triplicate but additional incubations were carried out across a number of analytical batches, giving greater replication and a greater understanding of system variability. Microsomal scaling factors used for human liver microsome incubations in the literature vary. Intrinsic clearance and predicted in vivo clearance rates calculated using alternative scaling factors found in the literature is provided in Appendix A, to aid comparison of data between publications.

### 3.8. Metabolic Stability—Pooled Human Cryopreserved Hepatocyte Incubations

Test compounds and positive controls (verapamil, 7-ethoxycoumarin (7-EC) and 7-hydroxycoumarin (7-HC)) were incubated at 0.5 µM in pooled human cryopreserved hepatocytes (pHHeps, 50 donor pool, 0.25 million cells mL^−1^). Total incubation volume was 400 µL. Procedure: Cells were thawed and added to cryopreserved hepatocyte recovery media, centrifuged (100× *g*, 10 min, 22 °C), counted and diluted in WME (37 °C, bubbled with 5% CO_2_:95% O_2_). To initiate reactions, 200 µL of drug working solution (1 µM) was added to 200 µL cell suspension (0.5 million cells mL^−1^). Samples were mixed and 20 µL sampled immediately and added to 80 µL ACN containing donepezil (IS, 50 ng mL^−1^) in a 96 deep-well 2 mL plate. Plates were incubated at 37 °C with 5% CO_2_ with agitation at 95 rpm. Further aliquots were taken at 3, 6, 9, 15, 30, 45, 60 min. Samples were diluted with 100 µL of water/ACN (80:20 *v*/*v*) and centrifuged (2800 rpm, 10 min, 22 °C) prior to UPLC-MS/MS analysis. Each SCRA was incubated at least in triplicate but additional incubations were carried out across a number of analytical batches, giving greater replication and a greater understanding of system variability. Differing values for hepatocyte cell density are reported in the literature. Intrinsic clearance and predicted in vivo clearance rates calculated using an alternative hepatocyte cell density is provided in Appendix A, to aid comparison of data between publications.

Rate constants (*k*, min^−1^) and half-lives (t_1/2_) were calculated using XLfit 5.3.1 add-in (IDBS, Surrey, UK) for Microsoft Excel 2013, version 15.0 (Microsoft, Redmond, WA, US), calculated from plots of analyte/IS peak area ratio against time. Microsomal intrinsic clearance (CL_int micr_) (pHLM data) and intrinsic clearance (CL_int_) scaled to whole-liver dimensions for humans (pHLM and pHHeps data) were calculated from *k*, liver and body weight estimates and scaling factors [13,96,116]. Hepatic clearance (CL_H_) and extraction ratios (E_H_) were estimated based on the corresponding human CL_int_ values, fraction unbound and estimates of liver blood flow rate (human 21 mL min^−1^ kg^−1^) [96,117,118].

The equations used for in vitro intrinsic clearance calculations and human in vivo clearance estimation were as described by Baranczewski et al. (2006) [96], Rane, Wilkinson & Shand (1977) [117] and Obach et al. (1997) [118] and are provided in Appendix A.

## 4. Conclusions

Many pharmacokinetic studies on SCRAs investigate metabolite formation as an aid to toxicological analysis and interpretation. Determination of intrinsic clearance of SCRAs by pHLM is also relatively common. However, few investigate SCRA clearance by HHeps or account for other pharmacokinetic parameters such as plasma stability or protein binding, or compare the clearance rates for a range of SCRAs of closely related structural classes in the same study. Here, for the first time, human in vivo clearance predictions are provided for valinate and *tert*-leucinate indole- and indazole-3-carboxamide SCRAs based on experimentally derived PPB data and in vitro intrinsic clearance by both pHLM and pHHeps. These compounds are lipophilic and highly protein bound, and so the parent compounds of many may only be present in authentic urine samples in small proportions compared to their metabolites. The results presented here support reports that, although some of these compounds are relatively unstable in vitro, as they are highly plasma protein bound the parent drug is likely stored in the adipose tissue of chronic users and may be detectable for an extended duration in plasma and urine following redistribution.

Overall, the tested SCRAs were cleared rapidly in vitro in both pHLM and pHHeps; (*S*)- enantiomers were cleared at a faster rate than (*R*)- enantiomers, except for compounds containing an alkene tail group when incubated in pHLM. Between compounds, the head group of the molecule is the one of the most important rate-determining factors, with valine (dimethyl) methyl ester (AMB-) compounds cleared the fastest, followed by *tert*-leucine (trimethyl) methyl ester (MDMB-) compounds, and with valinamide (dimethyl) (AB-) compounds cleared at the overall slowest rate. Once PPB data were considered, predicted in vivo hepatic clearance rates were much slower than in vitro intrinsic clearance. Despite some observed differences in intrinsic clearance between enantiomers, once protein binding was accounted for predicted whole-body clearance rates and hepatic extraction ratios were comparable between enantiomers. Predicted in vivo rates however varied widely between compounds suggesting that these compounds will be present in the circulatory system for varied durations, and thus may have varied durations of action, creating uncertainty for users where the specific drug as well as dose present is unknown.

## Figures and Tables

**Figure 1 molecules-26-01396-f001:**
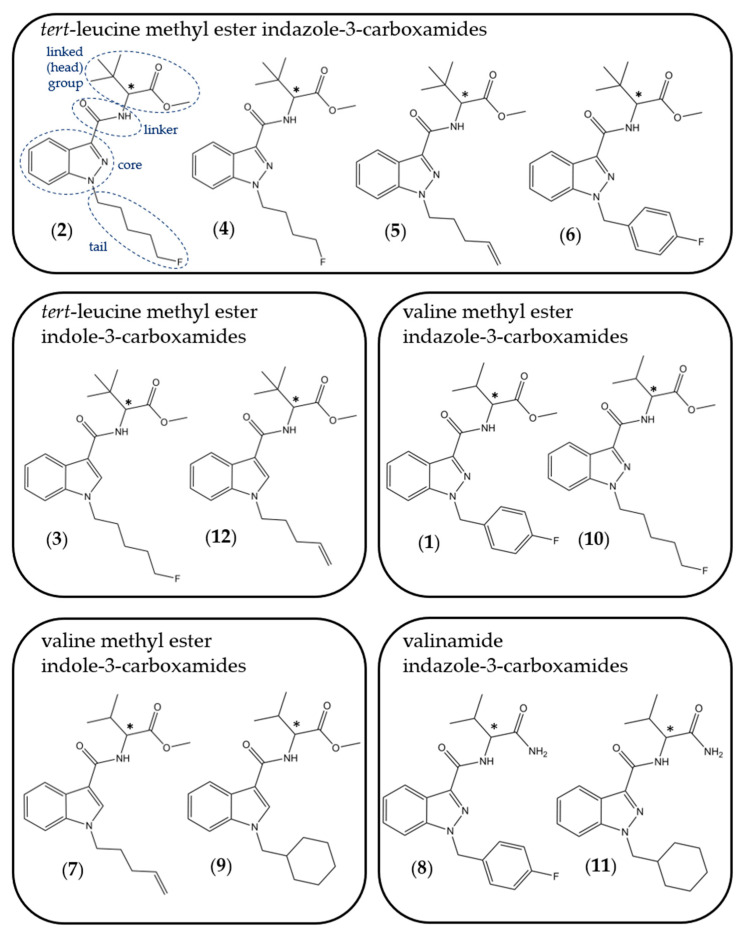
Structures of synthetic cannabinoid receptor agonists (SCRAs) involved in this study. (**1**) AMB-FUBINACA; (**2**) 5F-MDMB-PINACA; (**3**) 5F-MDMB-PICA; (**4**) 4F-MDMB-BINACA; (**5**) MDMB-4en-PINACA; (**6**) MDMB-FUBINACA; (**7**) AMB-4en-PICA; (**8**) AB-FUBINACA; (**9**) AMB-CHMICA; (**10**) 5F-AMB-PINACA; (**11**) AB-CHMINACA; (**12**) MDMB-4en-PICA. Asterisk (*) indicates chiral centre. Numbering used in the Figure reflects the order in which structures are mentioned in the text.

**Figure 2 molecules-26-01396-f002:**
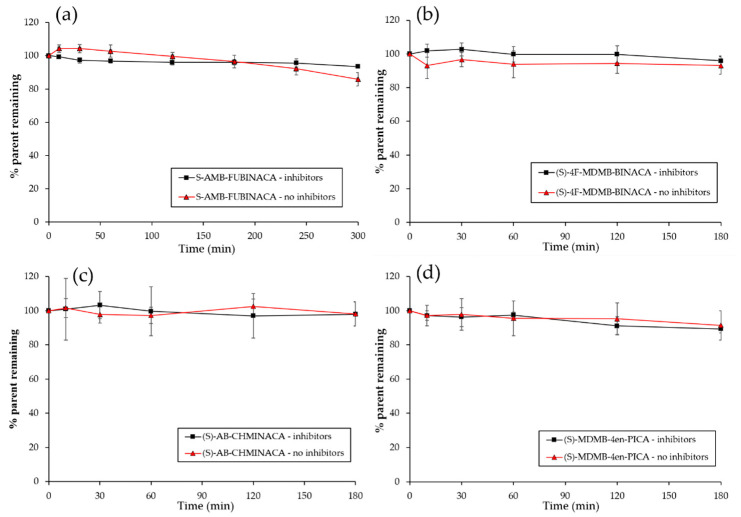
Stability of SCRAs (**a**) (*S*)-AMB-FUBINACA (**1**); (**b**) (*S*)-4F-MDMB-BINACA (**4**); (**c**) (*S*)-AB-CHMINACA (**11**); (**d**) (*S*)-MDMB-4en-PICA (**12**) in pooled human plasma in the presence and absence of esterase enzyme (*n* ≥ 3 ± SD). Full data in Appendix A.

**Figure 3 molecules-26-01396-f003:**
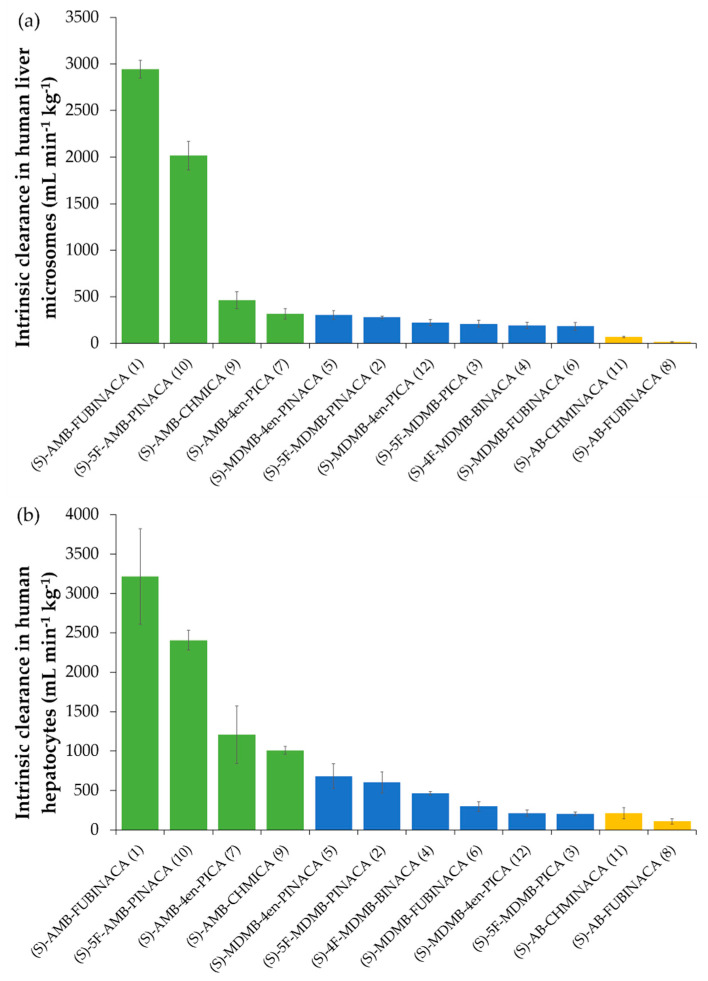
Intrinsic clearance (CL_int_, mL min^−1^ kg^−1^) of SCRAs ((*S*)-enantiomers) in (**a**) pooled human liver microsome incubations (*n* ≥ 3 ± SD); (**b**) pooled cryopreserved human hepatocyte incubations (*n* ≥ 3 ± SD).

**Figure 4 molecules-26-01396-f004:**
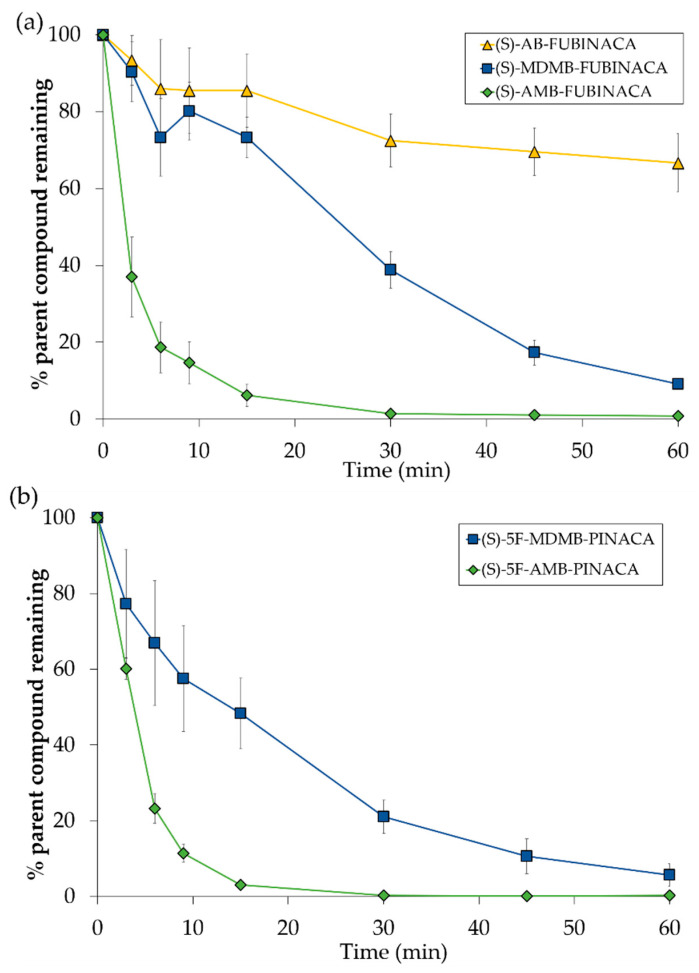
(**a**) Differences in clearance of (*S*)-AB-FUBINACA (**8**), (*S*)-MDMB-FUBINACA (**6**) and (*S*)-AMB-FUBINACA (**1**) incubated in pooled human cryopreserved hepatocytes (pHHeps) (*n* ≥ 3 ± SD). (**b**) Differences in clearance of (*S*)-5F-MDMB-PINACA (**2**) and (*S*)-5F-AMB-PINACA (**10**) incubated in pooled human cryopreserved hepatocytes (pHHeps) (*n* ≥ 3 ± SD).

**Table 1 molecules-26-01396-t001:** Predicted Log P and experimentally derived Log D_7.4_ values for test SCRAs calculated using a chromatographic hydrophobicity index (CHI). SwissADME data are shown, with log P ranges obtained using Gastroplus, MoKa, Canvas and XlogP software provided for comparison (see Appendix A for full data set). Data for the azaindole-3-carboxamide SCRA, 5F-AB-P7AICA are shown for comparative purposes.

Compound	Experimental Log D_7.4_	SwissADME Predicted Log P	Log P Range from Other Software Packages
MDMB-4en-PINACA (**5**)	4.95	3.41	3.61–4.00
AMB-CHMICA (**9**)	4.77	3.84	4.30–5.51
MDMB-FUBINACA (**6**)	4.69	3.83	4.08–4.24
5F-MDMB-PINACA (**2**)	4.50	3.63	3.76–3.90
MDMB-4en-PICA (**12**)	4.40	3.77	3.98–4.98
AMB-FUBINACA (**1**)	4.28	3.50	3.75–4.09
4F-MDMB-BINACA (**4**)	4.18	3.39	3.33–3.40
5F-AMB-PINACA (**10**)	4.07	3.41	3.31–3.64
5F-MDMB-PICA (**3**)	4.06	3.98	4.10–4.90
AMB-4en-PICA (**7**)	3.97	3.53	3.50–4.83
AB-CHMINACA (**11**)	3.71	2.91	3.10–3.55
AB-FUBINACA (**8**)	2.81	2.80	2.66–3.24
5F-AB-P7AICA	-	2.45	1.57–2.74

**Table 2 molecules-26-01396-t002:** Short-term plasma stability of tested SCRAs and positive control (procaine). ^a^ indicates data for 5 h incubations.

Compound	Parent Compound Remaining (%) (3 h; ^a^ 5 h)			
	No Esterase Inhibitors	*n* =	Esterase Inhibitors	*n* =
Procaine (control)	0.03 ± 0.02; ^a^ 0.08 ± 0.11	5	98.3 ± 8.0; ^a^ 91.0 ± 1.7	5
(*S*)-AB-FUBINACA (**8**)	104.4 ± 1.1	3	102.7 ± 2.5	3
(*S*)-AB-CHMINACA (**11**)	98.2 ± 7.2	3	97.9 ± 7.0	3
(*S*)-5F-MDMB-PICA (**3**)	97.3 ± 6.4	3	99.5 ± 5.8	3
(*R*)-5F-MDMB-PICA (**3**)	103.2 ± 14.5	3	97.7 ± 5.6	3
(*S*)-AMB-FUBINACA (**1**)	96.6 ± 3.8; ^a^ 85.9 ± 4.0	4	96.1 ± 1.6; ^a^ 93.5 ± 0.1	3
(*S*)-MDMB-4en-PINACA (**5**)	94.8 ± 0.7	3	89.0 ± 3.3	3
(*R*)-MDMB-4en-PINACA (**5**)	87.2 ± 6.9	3	89.4 ± 3.4	3
(*S*)-MDMB-FUBINACA (**6**)	94.2 ± 6.3	3	91.6 ± 2.2	3
(*S*)-AMB-CHMICA (**9**)	93.9 ± 9.3	3	90.5 ± 3.4	3
(*S*)-4F-MDMB-BINACA (**4**)	93.1 ± 5.1	3	96.0 ± 2.9	3
(*R*)-4F-MDMB-BINACA (**4**)	91.1 ± 6.8	3	96.0 ± 1.7	3
(*S*)-5F-AMB-PINACA (**10**)	93.0 ± 6.1	3	89.8 ± 1.5	3
(*S*)-AMB-4en-PICA (**7**)	92.6 ± 8.3	3	95.6 ± 6.8	3
(*R*)-AMB-4en-PICA (**7**)	91.7 ± 1.9	3	95.5 ± 2.9	3
(*S*)-5F-MDMB-PINACA (**2**)	92.5 ± 10.2	7	97.0 ± 1.1	3
(*R*)-5F-MDMB-PINACA (**2**)	95.9 ± 6.7	6	100.1 ± 7.0	7
(*S*)-MDMB-4en-PICA (**12**)	91.5 ± 8.6	3	89.4 ± 2.3	3
(*R*)-MDMB-4en-PICA (**12**)	90.3 ± 5.4	3	87.7 ± 11.0	3

**Table 3 molecules-26-01396-t003:** Plasma protein binding (PPB) of tested SCRA enantiomers.

Compound	PPB (%)	Fraction Unbound (fu)	*n* =
(*S*)-MDMB-FUBINACA (**6**)	99.5 ± 0.08	0.005 ± 0.0008	3
(*S*)-MDMB-4en-PINACA (**5**)	99.0 ± 0.01	0.010 ± 0.0001	3
(*R*)-MDMB-4en-PINACA (**5**)	98.1 ± 0.71	0.019 ± 0.0071	3
(*S*)-AMB-CHMICA (**9**)	98.8 ± 0.06	0.012 ± 0.0006	3
(*S*)-AMB-FUBINACA (**1**)	98.1 ± 0.08	0.019 ± 0.0008	3
(*S*)-AB-FUBINACA (**8**)	97.9 ± 0.44	0.021 ± 0.0044	3
(*S*)-5F-MDMB-PINACA (**2**)	97.8 ± 0.19	0.022 ± 0.0019	3
(*R*)-5F-MDMB-PINACA (**2**)	96.0 ± 0.58	0.040 ± 0.0058	4
(*S*)-AB-CHMINACA (**11**)	97.2 ± 2.19	0.028 ± 0.0219	3
(*S*)-MDMB-4en-PICA (**12**)	96.5 ± 0.32	0.035 ± 0.0031	4
(*R*)-MDMB-4en-PICA (**12**)	94.7 ± 1.11	0.053 ± 0.0111	4
(*S*)-AMB-4en-PICA (**7**)	94.7 ± 0.37	0.053 ± 0.0037	3
(*R*)-AMB-4en-PICA (**7**)	94.1 ± 0.10	0.059 ± 0.0010	3
(*S*)-5F-AMB-PINACA (**10**)	94.2 ± 0.08	0.058 ± 0.0008	3
(*S*)-4F-MDMB-BINACA (**4**)	93.9 ± 0.28	0.061 ± 0.0028	3
(*R*)-4F-MDMB-BINACA (**4**)	88.9 ± 0.49	0.111 ± 0.0049	4
(*S*)-5F-MDMB-PICA (**3**)	93.8 ± 0.07	0.062 ± 0.0007	3
(*R*)-5F-MDMB-PICA (**3**)	93.8 ± 0.15	0.062 ± 0.0015	3

**Table 4 molecules-26-01396-t004:** In vitro half-life, microsomal clearance (CL_int micr_) and intrinsic clearance (CL_int_) with predicted in vivo hepatic clearance (CL_H_) and hepatic extraction ratio (E_H_) for pooled human liver microsome incubations (*n* ≥ 3 ± SD). As plasma protein values are not available for some SCRA (*R*)-enantiomers, hepatic clearance and hepatic extraction ratio values have not been calculated.

Compound	T_1/2_ (min)	Microsomal Intrinsic Clearance, CL_int micr_ (mL min^−1^ mg Microsomal Protein^−1^)	Intrinsic Clearance, CL_int_ (mL min^−1^ kg^−1^)	Predicted In Vivo Hepatic Clearance, CL_H_ (mL min^−1^ kg^−1^)	Hepatic Extraction Ratio, E_H_	*n*
(*S*)-AMB-FUBINACA (**1**)	0.6 ± 0.02	2.182 ± 0.071	2944 ± 95.9	15.27 ± 0.14	0.73 ± 0.006	3
(*R*)-AMB-FUBINACA (**1**)	5.9 ± 0.48	0.237 ± 0.020	320 ± 26.8	-	-	6
(*S*)-5F-AMB-PINACA (**10**)	0.9 ± 0.07	1.494 ± 0.115	2016 ± 155.7	17.79 ± 0.20	0.85 ± 0.010	6
(*R*)-5F-AMB-PINACA (**10**)	6.5 ± 0.42	0.213 ± 0.014	288 ± 19.1	-	-	6
(*S*)-AMB-CHMICA (**9**)	4.1 ± 0.75	0.343 ± 0.069	463 ± 92.9	4.37 ± 0.68	0.21 ± 0.032	3
(*R*)-AMB-CHMICA (**9**)	8.4 ± 1.0	0.167 ± 0.019	226 ± 25.4	-	-	3
(*S*)-AMB-4en-PICA (**7**)	6.0 ± 1.0	0.236 ± 0.042	318 ± 56.6	9.29 ± 0.89	0.44 ± 0.042	10
(*R*)-AMB-4en-PICA (**7**)	4.1 ± 0.75	0.348 ± 0.068	469 ± 92.2	11.8 ± 0.97	0.56 ± 0.046	11
(*S*)-MDMB-4en-PINACA (**5**)	6.3 ± 1.0	0.226 ± 0.037	305 ± 49.3	2.66 ± 0.37	0.13 ± 0.018	7
(*R*)-MDMB-4en-PINACA (**5**)	5.5 ± 0.59	0.253 ± 0.026	341 ± 34.6	4.94 ± 0.39	0.24 ± 0.019	7
(*S*)-5F-MDMB-PINACA (**2**)	6.7 ± 0.31	0.207 ± 0.009	280 ± 12.8	4.76 ± 0.17	0.23 ± 0.008	5
(*R*)-5F-MDMB-PINACA (**2**)	8.0 ± 1.1	0.176 ± 0.024	237 ± 31.7	6.52 ± 0.61	0.31 ± 0.029	5
(*S*)-5F-MDMB-PICA (**3**)	9.0 ± 1.4	0.156 ± 0.027	211 ± 36.9	8.01 ± 0.84	0.38 ± 0.04	3
(*R*)-5F-MDMB-PICA (**3**)	11 ± 0.11	0.132 ± 0.001	178 ± 1.86	7.22 ± 0.05	0.34 ± 0.002	3
(*S*)-MDMB-4en-PICA (**12**)	8.5 ± 1.3	0.166 ± 0.024	224 ± 32.4	5.69 ± 0.61	0.27 ± 0.029	11
(*R*)-MDMB-4en-PICA (**12**)	5.4 ± 0.74	0.260 ± 0.040	350 ± 53.3	9.81 ± 0.75	0.47 ± 0.036	9
(*S*)-4F-MDMB-BINACA (**4**)	9.9 ± 1.9	0.143 ± 0.025	193 ± 33.8	7.51 ± 0.88	0.36 ± 0.042	3
(*R*)-4F-MDMB-BINACA (**4**)	14 ± 0.68	0.096 ± 0.005	130 ± 6.26	8.55 ± 0.24	0.41 ± 0.012	3
(*S*)-MDMB-FUBINACA (**6**)	11 ± 2.4	0.135 ± 0.030	183 ± 40.2	0.87 ± 0.18	0.04 ± 0.009	6
(*R*)-MDMB-FUBINACA (**6**)	20 ± 2.5	0.071 ± 0.009	95.4 ± 12.3	-	-	6
(*S*)-AB-CHMINACA (**11**)	27 ± 2.9	0.051 ± 0.005	69.1 ± 7.35	1.77 ± 0.17	0.08 ± 0.008	3
(*R*)-AB-CHMINACA (**11**)	43 ± 5.0	0.033 ± 0.004	44.1 ± 4.89	-	-	3
(*S*)-AB-FUBINACA (**8**)	118 ± 28	0.012 ± 0.003	16.4 ± 4.24	0.34 ± 0.09	0.02 ± 0.004	3
(*R*)-AB-FUBINACA (**8**)	145 ± 45	0.010 ± 0.003	13.7 ± 4.06	-	-	3

**Table 5 molecules-26-01396-t005:** In vitro half-life and intrinsic clearance (CL_int_) with predicted in vivo hepatic clearance (CL_H_) and hepatic extraction ratio (E_H_) for pooled human cryopreserved hepatocyte incubations (*n* ≥ 3 ± SD). As plasma protein values are not available for some SCRA (*R*)-enantiomers, hepatic clearance and hepatic extraction ratio values have not been calculated.

Compound	T_1/2_ (min)	Intrinsic Clearance, CL_int_ (mL min^−1^ kg^−1^)	Predicted In Vivo Hepatic Clearance, CL_H_ (mL min^−1^ kg^−1^)	Hepatic Extraction Ratio, E_H_	*n*
(*S*)-AMB-FUBINACA (**1**)	2.5 ± 0.55	3216 ± 607	15.52 ± 0.85	0.74 ± 0.041	8
(*R*)-AMB-FUBINACA (**1**)	3.1 ± 0.57	2557 ± 463	-	-	8
(*S*)-5F-AMB-PINACA (**10**)	3.2 ± 0.16	2404 ± 125	18.25 ± 0.12	0.87 ± 0.006	3
(*R*)-5F-AMB-PINACA (**10**)	4.7 ± 0.26	1623 ± 87.0	-	-	3
(*S*)-AMB-4en-PICA (**7**)	6.9 ± 2.1	1205 ± 363	15.56 ± 1.24	0.74 ± 0.059	8
(*R*)-AMB-4en-PICA (**7**)	5.5 ± 2.1	1589 ± 616	16.79 ± 1.33	0.80 ± 0.063	8
(*S*)-AMB-CHMICA (**9**)	7.6 ± 0.35	1011 ± 48.5	7.68 ± 0.22	0.37 ± 0.011	3
(*R*)-AMB-CHMICA (**9**)	8.3 ± 0.35	918 ± 38.2	-	-	3
(*S*)-MDMB-4en-PINACA (**5**)	12 ± 2.6	683 ± 158	5.11 ± 0.89	0.24 ± 0.042	7
(*R*)-MDMB-4en-PINACA (**5**)	22 ± 3.6	363 ± 66.5	5.16 ± 0.70	0.25 ± 0.033	7
(*S*)-5F-MDMB-PINACA (**2**)	13 ± 3.3	604 ± 135	8.05 ± 1.15	0.38 ± 0.055	12
(*R*)-5F-MDMB-PINACA (**2**)	23 ± 1.8	334 ± 25.5	8.16 ± 0.38	0.39 ± 0.018	6
(*S*)-4F-MDMB-BINACA (**4**)	16 ± 0.74	466 ± 21.6	12.08 ± 0.23	0.58 ± 0.011	3
(*R*)-4F-MDMB-BINACA (**4**)	58 ± 17.2	139 ± 37.4	8.77 ± 1.43	0.42 ± 0.068	4
(*S*)-MDMB-FUBINACA (**6**)	26 ± 5.0	298 ± 62.7	1.39 ± 0.27	0.07 ± 0.013	3
(*R*)-MDMB-FUBINACA (**6**)	32 ± 5.4	239 ± 40.7	-	-	3
(*S*)-MDMB-4en-PICA (**12**)	37 ± 6.9	213 ± 41.6	5.46 ± 0.78	0.26 ± 0.037	7
(*R*)-MDMB-4en-PICA (**12**)	44 ± 6.3	176 ± 24.0	6.42 ± 0.62	0.31 ± 0.029	7
(*S*)-5F-MDMB-PICA (**3**)	38 ± 4.6	204 ± 23.2	7.87 ± 0.58	0.37 ± 0.027	3
(*R*)-5F-MDMB-PICA (**3**)	54 ± 9.4	145 ± 24.1	6.27 ± 0.74	0.30 ± 0.035	3
(*S*)-AB-CHMINACA (**11**)	39 ± 8.8	210 ± 72.9	4.52 ± 1.12	0.22 ± 0.053	6
(*R*)-AB-CHMINACA (**11**)	38 ± 8.7	213 ± 54.3	-	-	5
(*S*)-AB-FUBINACA (**8**)	76 ± 24	110 ± 34.5	2.06 ± 0.58	0.10 ± 0.028	8
(*R*)-AB-FUBINACA (**8**)	69 ± 11	113 ± 15.6	-	-	8

## Data Availability

Data are available in a publicly accessible repository: Brandon, A.M.; Antonides, L.H.; Riley, J.; Epemolu, O.; Read, K.D.; McKenzie, C. 2021. Supporting data for “A Systematic Study of the In Vitro Pharmacokinetics and Estimated Human In Vivo Clearance of Indole and Indazole-3-Carboxamide Synthetic Cannabinoid Receptor Agonists Detected on the Illicit Drug Market”; available at doi:10.15132/10000164.

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
