# Peer review of "A Systematic Study of the In Vitro Pharmacokinetics and Estimated Human In Vivo Clearance of Indole and Indazole-3-Carboxamide Synthetic Cannabinoid Receptor Agonists Detected on the Illicit Drug Market"

_molecules, 2021, doi:10.3390/molecules26051396_

Round 1
Reviewer 1 Report
The manuscript is focus on the study of the pharmacokinetics properties of some synthetic cannabinoid receptor agonists in order to investigate their physicochemical parameters and structure-metabolism relationships. In recent years, the use of synthetic cannabinoids as drugs of abuse has been reported. Thus, it is relevant to know how this kind of drugs are distributed and metabolized in the organism to know its toxicological implications.
Minor:
In Figure 1 it is not clear which subunit corresponds to the head.
It is not clear why the authors use the numbers in the order shown in Figure 1.
The last paragraph of Introduction is written in future, but the study is already done. It should be written in past tense.
Author Response
We thank the reviewer for the time they have taken to review the submitted manuscript.
- Response to Reviewer 1 comments
Comment 1: In Figure 1 it is not clear which subunit corresponds to the head.
Response 1: Figure 1 was updated to clarify that the ‘head’ group refers to the linked group by indicating linked (head) group
Comment 2: It is not clear why the authors use the numbers in the order shown in Figure 1.
Response 2: Figure 1 legend has been updated to indicate that numbering reflects the order in which structures are mentioned in the text.
Comment 3: The last paragraph of Introduction is written in future, but the study is already done. It should be written in past tense.
Response: We apologise for this oversight. The last paragraph of the introduction (L157-L169) has been updated to past tense.
Reviewer 2 Report
This paper reports in vitro pharmacokinetics and estimated human in vivo clearance of selected synthetic cannabinoid receptor agonists. SCRAs are heterogeneous compounds originally intended as probes of the endogenous cannabinoid system, but also as potential therapeutic agents. The authors take up a very important issue of SCRAs compounds detection into herbal materials, papers, and e-liquids for smoking and vaping, not only on the illicit market as pure substances. It is crucial to understand the physicochemical parameters underpinning their behavior in the biological system, as well as their pharmacokinetics. This paper help to understand the factors that influence the pharmacokinetics and metabolism of valinate or tert-leucinate indole- and indazole-3-carboxamide SCRAs. The manuscript is well-written, introduction as well as results and discussion sections clearly present the interesting results, the conclusions are supported by the results. Thus I recommend this manuscript to publish in the present form.
Author Response
We thank the reviewer for the time taken to review the submitted manuscript.
There were non comments to address.
Reviewer 3 Report
Brandon and colleagues' manuscript is interesting. The authors characterize new classes of synthetic cannabinoids in in-silico models. Considering that characterizing new molecules that act on the endocannabinoid system is very important for the understanding of many pathologies, the manuscript addresses a hot-topic. However, the experimental procedures are not clear and some concerns should be clarified by the authors.
1) The title reports “in vivo characterization”. However, no drug treatment on murine models was performed. This is a severe shortage of the manuscript.
2) The in-vitro experiments are not clear. In vitro studies assume the use of at least cell lines. The authors should clarify these aspects.
3) The authors use human plasma for some experiments. I wonder who the subjects who donated the plasma are. What are the epidemiological characteristics of human subjects? Are they controls? Do they have pathologies? These are important details.
4) The greatest criticality concerns the absence of statistical analysis. What statistical survey was performed?
5) I suggest to the authors a thorough revision of the manuscript and I suggest the integration of these manuscripts:
Manca I et al. Novel pyrazole derivatives as neutral CB₁ antagonists with significant activity towards food intake. Eur J Med Chem. 2013
Lazzari P et al. Synthesis and pharmacological evaluation of novel 4-alkyl-5-thien-2'-yl pyrazole carboxamides. Cent Nerv Syst Agents Med Chem. 2012 Dec;12(4):254-76. doi: 10.2174/187152412803760636.
Mastinu A, Pira M, Pani L, Pinna GA, Lazzari P. NESS038C6, a novel selective CB1 antagonist agent with anti-obesity activity and improved molecular profile. Behav Brain Res. 2012 Oct 1;234(2):192-204.
Author Response
We thank the reviewer for taking the time to review our manuscript. The comments were helpful and we considered our responses to the points raised carefully.
Comment 1: The title reports “in vivo characterization”. However, no drug treatment on murine models was performed. This is a severe shortage of the manuscript.
Response 1: The authors highlight that at no point in the manuscript do they claim to have carried out “in vivo characterization” the title states “estimated human in vivo clearance” as in vivo clearance values provided were based on experimental in vitro data, published physiological parameters and well-established extrapolation models. The methods used for in vivo extrapolation used have been clearly and thoroughly explained in the manuscript and alternative calculations using slightly different parameters have been included in the supplementary information to allow comparison between the parameters commonly used in Forensic Toxicology-related publications and those used in industry in pharmaceutical DMPK models, both professional contexts in which the authors have extensive experience. The purpose of the in vivo extrapolations detailed repeatedly stated: to link to recently published data suggesting long detection times for these compounds in toxicological samples, despite them having fast in vitro clearance using human liver microsomes and human hepatocytes. This is clearly one of the novel aspects of the submitted paper as using hepatic in vitro clearance data without the inclusion of plasma protein binding data clearly vastly underestimates in vivo clearance. The use of animal/murine models in this instance is not appropriate due to known significant differences in expression of, in particular, carboxylesterase enzymes, in both plasma and liver of some animals used in vivo testing, which are particularly important in the metabolism of the methyl ester and amide containing SCRAs detailed in the study.
Comment 2: The in-vitro experiments are not clear. In vitro studies assume the use of at least cell lines. The authors should clarify these aspects.
Response 2: In vitro intrinsic clearance studies in this work involved incubation of test compounds with both pooled human liver microsomes and cryopreserved human hepatocyte cells, both of which are standard and accepted techniques in both academic and pharmaceutical drug development research related to drug metabolism and pharmacokinetic (DMPK) studies. The experimental information provided is exactly in line with that provided in hundreds of other peer-reviewed drug metabolism studies using human liver microsomes and human hepatocytes both from a pharmaceutical DMPK context and in published forensic toxicology related metabolite identification studies, many of which are cited in the extensive reference list of the submitted manuscript. We would ask the reviewer to clarify their request for further information - we have provided the source and nature of the human liver microsomes and human hepatocytes used in detail and provided a detailed experimental description to allow the research to be replicated. This is provided in sections 3.7 and 3.8, respectively. The last paragraph of the introduction has been updated to clarify that this aspect of the work involved in vitro incubations, as has the first sentence of section 2.4.
Comment 3: The authors use human plasma for some experiments. I wonder who the subjects who donated the plasma are. What are the epidemiological characteristics of human subjects? Are they controls? Do they have pathologies? These are important details.
Response 3: Thank you for this comment and we are very happy to provide further information on the source of human plasma used which we agree was somewhat lacking. We have provided additonal information in the supplementary information (section 3) and additional detail has been added to the main manuscript at section 3.8 L608-L613: “The pooled plasma used for both plasma stability and plasma binding studies originated from 50 donors, collected as whole blood from donors in the United States at an FDA-approved collection center, processed into plasma by centrifugation and immediately frozen (Innovative Research; Novi, Michigan, United States) and purchased from Patricell (Nottingham, UK). Plasma was tested by the manufacturers for a range of FDA-required viral markers."
Comment: 4) The greatest criticality concerns the absence of statistical analysis. What statistical survey was performed?
Response: The nature of the statistical tests requested by the reviewer have not specified nor has their proposed purpose. The authors have presented trend data providing fundamental data on the intrinsic clearance of structurally related substances using the same human liver microsomes and human hepatocytes for each substance tested for comparability. This has been carried out specifically to further understanding of structure metabolism relationships for these highly prevalent and harmful illicit drugs and takes into account that their structures are constantly evolving in response to legislative/detectabilitychanges in the SCRA drug market. We see little value in stating that the clearance rate of one compound is statistically different from another as compounds vary in the combination of structural subunits. The paper sets our a series of structure metabolism relationships and seeks to explain why some types of compounds have different detection windows. e.g. amides in the head/linked group slows down metabolism especially if there is another even more metabolically resistant tail group e.g. the 4-fluorobenzyl group in AB-FUBINACA. We also showed the apparent affect of steric hindrance on enzymatic (carboxylesterase) attack on the head/linker ester/amide e.g. AMB-FUBINACA with a dimethyl moiety vs MDMB-FUBINACA with a trimethyl moiety. Determining if one compound is cleared at a statistically significant faster rate than another has little relevance to the real-world application of the data to aid interpretation of results in forensic toxicology casework. What we have clearly identified is that there is a clear structure metabolism relationship related to between class structural characteristics but that within those classes there is also a range of structural metabolism relationships related to the presence of the different structural subunits, some of which are resistant to metabolism (e.g. a fluorobenzyl tail), others which are less resistant to enzymatic metabolism (e.g. a cyclohexyl or alkene tail). This can then be applied to emerging compounds as they appear (for example 4F-MDMB-BICA; 5F-EMB-PICA and ADB-4en-PINACA have all very recently been detected). The study has not been designed to carry out statistical significance testing between analogues and therefore would likely be statistically underpowered and unbalanced. To carry out such statistical data analysis as suggested in a post-hoc manner for a purpose to which the study was not designed, and would, in our opinion, be scientifically incorrect, of little applied use and potentially misleading. The authors feel that statistical analysis is not required for the experiments included in this manuscript and refer the reader to similar studies referenced in Table 6 of the manuscript, none of which included any of the statistical testing requested by the reviewer in this case.
Now that we have identified the principal features of the structure metabolism relationship as reported in this study, if we were to do a follow up study with that specific purpose of statistically testing the differences in clearance rates, we would limit the SCRA class subunit types so that compounds were the most different e.g. AB-FUBINACA vs AMB-FUBINACA increased replications and had correctly powered statistical tests. It is clear from the data that these two compounds have vastly different clearance rates and there is no logical reason to apply an underpowered statistical test to prove it. Additionally, of course this does not take into account all of the natural inter-individual variation in enzyme expression that would be found in users of the drugs. The paper sets out to identify the fundamental structure metabolism relationship 'rules'. and then applies this knowledge to estimate how that would translate into a real life in vivo situation - a particular novelty of the submitted manuscript compared to previous publications of this type.
Comment 5: I suggest to the authors a thorough revision of the manuscript and I suggest the integration of these manuscripts:
Response: With all due respect, although the suggested citations refer to excellent and interesting work, the authors feel stronly that the papers suggested for inclusion are not relevant to the submitted paper under review and therefore we have chosen not to include reference to them in the revised manuscript.
The paper submitted for review focusses on proven potent indole and indazole-3-carboxamide synthetic cannabinoid receptor agonists which either currently or have recently circulated on the illicit market, and their closely related structural analogues included to help unravel structure metabolism relationships.
Specifically, in terms of the studies we have been asked to cite/integrate into the study under review:
Manca I et al. Novel pyrazole derivatives as neutral CB₁ antagonists with significant activity towards food intake. Eur J Med Chem. 2013
The pyrazole-derivatives (some of which have a carboxamide linker) referred to in the paper are interesting but are structurally unrelated to the substances in the paper submitted for review. They have not (to date) been detected on the illicit drugs market as synthetic cannabinoid receptor agonists and are not of current toxicological interest in a forensic toxicology context, the principal focus of the study submitted for review. The compounds, particularly compound 1 as an example of a compound which binds selectively to CB1, may be of interest to the illicit market in future and we shall certainly add that to our watchlist, however these compounds are not relevant to the current study. Moreover, the study we have been requested to cite shows binding affinity to the CB1 receptor and not activation as measured by downstream signalling assays. ALL compounds cited in this study have been shown to both bind and activate the CB1 receptor using b-arrestin based functional activity bioassays (see the cited work of our group, Antonides et al., 2019; Antonides et al 2020 provided in the submitted paper's reference list, and the work of our collaborators the Stove Group at Ghent University, Belgium who developed the functional assays). There are many such compounds, such as those in the requested citation, which demonstate selective CB1 binding in the scientific literature and they also do not come into the scope of this work and are not cited.
Lazzari P et al. Synthesis and pharmacological evaluation of novel 4-alkyl-5-thien-2'-yl pyrazole carboxamides. Cent Nerv Syst Agents Med Chem. 2012 Dec;12(4):254-76. doi: 10.2174/187152412803760636.
We were not able to access the above-noted study as it is not open access and is behind a paywall at the publishers Bentham Science for which we have no subscription, from the abstract of the paper the same arguments noted for the previously discussed study hold for this paper and therefore it is not appropriate to cite it in the study under review.
Mastinu A, Pira M, Pani L, Pinna GA, Lazzari P. NESS038C6, a novel selective CB1 antagonist agent with anti-obesity activity and improved molecular profile. Behav Brain Res. 2012 Oct 1;234(2):192-204.
The paper we have been requested to cite refers to the CB1 agonist NESS038C6, which is a tricyclic core compound with selective CB1 binding activity (but downstream CB1 receptor activation has not been measured as far as we can find). Again, whilst an interesting compound and certainly of interest in the particular context of the study published (obesity not psycho-activity), it has never appeared on the illicit drug market. We therefore, respectfully, feel that it is inappropriate to cite the study in the paper submitted for review.
Round 2
Reviewer 3 Report
Although the authors have abundantly replied to my objections, important critical issues must be resolved.
1) the most important concerns the epidemiological aspects of biological samples of human origin used in the experiments. Each time a sample of human origin is used, it is necessary to evaluate the inclusion criteria (age, sex, diseases, etc.). It seems that the authors do not have this information. This problem must be solved otherwise their thesis collapses.
2) Looking at point 1, it is necessary to apply a statistical approach that highlights how different clinical phenotypes respond to illicit cannabinoids.
3) Finally, if the authors focus their attention on illicit cannabinoids, they should highlight this in the title. Unfortunately, the manuscript would have potential, but it needs further improvements before publication, including a bibliographic one.
Author Response
We thank the reviewer for their comments and request for further clarifications.
Comment 1: the most important concerns the epidemiological aspects of biological samples of human origin used in the experiments. Each time a sample of human origin is used, it is necessary to evaluate the inclusion criteria (age, sex, diseases, etc.). It seems that the authors do not have this information. This problem must be solved otherwise their thesis collapses.
Response 1:
We have addressed this comment on the following ways:
(i) We have now provided citations (refs 110-114) for the certificates of analysis (CoA) for all batches of pooled human liver microsomes and pooled human cryopreserved hepatocytes used in the study. The CoAs provide detailed information on the donors to each batch (n=50, 25 male and 25 female donors in each batch) as well as a range of other information. We believe this now addresses the concern raised and allows for independent verification of the donor pool for the in vitro platforms used in this study. This is more information than is usually provided in studies using such material and we are very happy to provide it. We have cited an additional reference in the text noting the sources of inter- and intra-individual variability in the text (reference 81). Reference numbering has been updated.
(ii) We have more clearly identified that human liver microsomes and human hepatocytes used in the study are pooled samples from 50 donors throughout the manuscript and now use the terms “pHLM” and “pHHeps” when referring to them with the ‘p’ indicating pooled. We now refer to the hepatocytes used consistently as pooled human cryopreserved hepatocytes.
(iii) We would like to reiterate that the use of pHLM and pHHeps is very common in metabolism studies of new psychoactive substances appearing on the illicit drug market. This is due to the ethical restrictions in place (for good reason) on in vivo testing of highly potent and often harmful illicit drugs in humans. Nearly all in vivo testing originates from toxicological samples from clinical or forensic toxicology samples where dosage taken and time intervals are unknown. In some previous studies the number of pooled donors is either not provided or is 10, considerably lower than the pools used in our study. We have been very transparent in the systems we have used and are using pHLM and pHHeps with larger donor pools to encapsulate greater variability in enzyme expression. We have added the following text to the manuscript to reiterate why we are using such pooled in vitro platforms and why we are not studying inter- and intra-individual variation:
“Each batch of pHLM and pHHeps used in the study comprised pooled biological material from 50 donors, 25 male and 25 female, of varying ages and health status. Details of the donors to each batch are provided in the certificate of analysis associated with each batch (see section 3.1 for further details of donors). It is recognized that there will be inter- and intra-individual variation in SCRA metabolism, due to age, sex, health status and polymorphisms [81]. The use of large donor pools for pHLM and pHHeps batches (n=50) reduces inter-batch variation and provides information on the fundamental structure metabolism relationships of SCRA analogues within and between structural classes and intrinsic clearance values produced relates to an ‘average individual’. This study does not set out to study differences in the ability of individuals, or phenotypes to metabolise SCRAs.”
We have also expanded the study limitations section:
“A limitation to this study, and indeed other research studies related to the metabolism of constantly emerging SCRAs and other NPS using pHLM and pHHeps, is that it does not consider the expected variation in metabolic capacity between individual users. The aim of the study is, however, to provide comparative information for the metabolism of structurally related SCRAs in an ‘average’ human using pooled donor in vitro platforms, to allow the deconvolution of the key structural features of The SCRAs that affect metabolic stability in isolation. The study provides a more accurate estimate of in vivo human hepatic clearance than previously reported by taking into account the effect of plasma protein binding. This is the first time such information has been incorporated into in vivo hepatic clearance calculations for the SCRA structural classes studied.”
Comment 2: Looking at point 1, it is necessary to apply a statistical approach that highlights how different clinical phenotypes respond to illicit cannabinoids.
Response 2: We believe that we have now addressed this comment in our previous response as much as we can. Whilst the study suggested is certainly an interesting one, it does not come under the scope of the study carried out and would be impossible to carry out retrospectively.
Comment 3: Finally, if the authors focus their attention on illicit cannabinoids, they should highlight this in the title. Unfortunately, the manuscript would have potential, but it needs further improvements before publication, including a bibliographic one.
Response 3: We have updated the title of the manuscript to “A systematic study of the in vitro pharmacokinetics and estimated human in vivo clearance of indole and indazole-3-carboxamide synthetic cannabinoid receptor agonists detected on the illicit drug market”.
Finally, in terms of the bibliography, this is an original research study with an extensive reference list (>118 references) all of which are relevant to the study described in the manuscript.
In addition to our responses to the specific comments above, we have made some further to the manuscript, updating the compound numbering to ensure it matched the order SCRAs are mentioned in the text (also in the supplementary information and all figures), ensured that alternative drug-market names and IUPAC names are provided the first time a SCRA is mentioned and reviewing the reference list to ensure that it complies fully to the journal referencing style. To incorporate the additional text noted in our response we have moved the positions of some of the Tables and Figures in the manuscript.
We thank the reviewer for their consideration of the submitted manuscript.